# Bacterial cyclic diguanylate signaling networks sense temperature

Henrik Almblad[1,9], Trevor E. Randall[1,9], Fanny Liu[1], Katherine Leblanc [1], Ryan A. Groves[1], Weerayuth Kittichotirat [2], Geoffrey L. Winsor [3], Nicolas Fournier[1], Emily Au[1], Julie Groizeleau[1], Jacquelyn D. Rich[1], Yuefei Lou[4], Elise Granton[4], Laura K. Jennings [5], Larissa A. Singletary[5], Tara M. L. Winstone[6], Nathan M. Good[5], Roger E. Bumgarner[5], Michael F. Hynes[1], Manu Singh [7], Maria Silvina Stietz[4], Fiona S. L. Brinkman[3], Ayush Kumar [7], Ann Karen Cornelia Brassinga[7], Matthew R. Parsek[5], Boo Shan Tseng [8], Ian A. Lewis [1], Bryan G. Yipp [4], Justin L. MacCallum[6] & Joe Jonathan Harrison [1✉]

Many bacteria use the second messenger cyclic diguanylate (c-di-GMP) to control motility, biofilm production and virulence. Here, we identify a thermosensory diguanylate cyclase (TdcA) that modulates temperature-dependent motility, biofilm development and virulence in the opportunistic pathogen *Pseudomonas aeruginosa*. TdcA synthesizes c-di-GMP with catalytic rates that increase more than a hundred-fold over a ten-degree Celsius change. Analyses using protein chimeras indicate that heat-sensing is mediated by a thermosensitive Per-Arnt-SIM (PAS) domain. TdcA homologs are widespread in sequence databases, and a distantly related, heterologously expressed homolog from the Betaproteobacteria order *Gallionellales* also displayed thermosensitive diguanylate cyclase activity. We propose, therefore, that thermotransduction is a conserved function of c-di-GMP signaling networks, and that thermosensitive catalysis of a second messenger constitutes a mechanism for thermal sensing in bacteria.

[1] Department of Biological Sciences, University of Calgary, Calgary, AB, Canada. [2] Systems Biology and Bioinformatics Research Group, Pilot Plant Development and Training Institute, King Mongkut's University of Technology Thonburi, Bangkok, Thailand. [3] Department of Molecular Biology and Biochemistry, Simon Fraser University, Burnaby, BC, Canada. [4] Cumming School of Medicine, University of Calgary, Calgary, AB, Canada. [5] Department of Microbiology, University of Washington, Seattle, WA, USA. [6] Department of Chemistry, University of Calgary, Calgary, AB, Canada. [7] Department of Microbiology, University of Manitoba, Winnipeg, MB, Canada. [8] School of Life Sciences, University of Nevada Las Vegas, Las Vegas, NV, USA. [9]These authors contributed equally: Henrik Almblad, Trevor E. Randall. ✉email: jjharris@ucalgary.ca

Temperature sets physical limits for life on Earth[1,2]. It affects the physiology and geographic distribution of organisms[3], including Eubacteria[4,5], across landscapes of all scales. Bacteria express a dynamic range of behaviors following temperature fluctuations, such as cold- or heat-shock responses to thermal extremes[6,7], taxis along thermal gradients[8], and differential expression of virulence factors at temperatures corresponding to those of endothermic hosts[9,10]. Unsurprisingly, bacteria have evolved many means for sensing temperature[9,10]. Known molecular heat-sensing strategies include RNA thermometers[10], temperature-sensitive transcription factors[11], and transmembrane proteins that initiate signal transduction in response to thermal changes in the thickness of lipid bilayers[12,13].

Cyclic diguanylate (c-di-GMP) signal transduction is thought to be integral to the control of diverse physiological and social behaviors among bacterial species from nearly every phylum[14]. C-di-GMP is enzymatically synthesized by diguanylate cyclases harboring glycine–glycine–aspartate–glutamate–phenylalanine (GGDEF) domains, and degraded by c-di-GMP-specific phosphodiesterases containing glutamate–alanine–leucine (EAL) or histidine–aspartate–glycine–tyrosine–proline (HD-GYP) domains[14–17]. Although c-di-GMP signaling has been studied since the 1980s[18], one key challenge to understanding it has been identifying the stimuli that activate diguanylate cyclases and c-di-GMP-specific phosphodiesterases[14]. Many of these proteins contain putative sensory domains adjacent to their GGDEF, EAL, or HD-GYP domains, yet the vast majority of these sensory domains have no known function.

Here we report the discovery of diguanylate cyclases in which a thermosensitive Per-Arnt-SIM domain is coupled with a GGDEF domain. We show that these proteins display high enzymatic rate–temperature dependencies that are analogous to those described for the cold- and- hot-sensing proteins of neurons[19–21]. We propose that these enzymes constitute molecular thermosensory devices that enable bacteria to calibrate intracellular c-di-GMP levels in response to temperature.

## Results

**Heat stimulates c-di-GMP biosynthesis.** In an effort to identify stimuli of bacterial c-di-GMP signal transduction networks, we exploited the diverse colony phenotypes displayed by different isolates of *Pseudomonas aeruginosa*. The variations in colony morphology of these isolates are often due to differences in their c-di-GMP signaling pathways[22–26]. A prime example of this phenomenon is the simultaneous isolation of smooth and rugose colony morphology variants from biofilms in laboratory bioreactors and chronic infections[26–31]. Although there may be exceptions, many smooth colony variants exhibit low intracellular c-di-GMP levels and reduced biofilm production, whereas rugose colony variants have high intracellular c-di-GMP levels and increased biofilm production[32].

Here, we used a paired collection of smooth and rugose *P. aeruginosa* colony variants previously isolated from the sputum of cystic fibrosis (CF) patients suffering from chronic lung infection[26,32]. The isolates of each pair were known to be close relatives by pulsed-field gel electrophoresis genotyping[26,32]. By varying the growth conditions, we found that we could change the colony morphology for a subset of these paired isolates such that one isolate of the pair would switch to the other colony phenotype. This tactic provided a platform for investigating gene–environment interactions affecting intracellular c-di-GMP levels. This approach led to the identification that one strain, *P. aeruginosa* CF39S, displayed temperature-dependent rugose colony morphology, whereas its close genetic relative, strain CF39, did not (Fig. 1a). Consistent with predictions from these

colony morphology phenotypes, *P. aeruginosa* CF39 displayed similar biofilm production at all tested temperatures, whereas CF39S had a capacity to produce biofilm that massively increased between 30 and 34 °C (Fig. 1b).

The literature is rife with descriptions that temperature regulates biofilm production in bacteria, yet connections between these observations and c-di-GMP signaling are limited[33,34]. We posited, therefore, that temperature-dependent increases in *P. aeruginosa* CF39S biofilm production might be correlated with increases in c-di-GMP. To test this hypothesis, nucleotide pools were extracted from cultures of *P. aeruginosa* CF39 and CF39S and c-di-GMP was measured using mass spectrometry. We observed that strain CF39S produced increasing quantities of c-di-GMP with respect to temperature in the physiologically relevant range of 28–41 °C (Fig. 1c). These data suggest that cellular components of *P. aeruginosa* CF39S behave like a molecular thermostat, responding to changes in temperature to fine-tune c-di-GMP synthesis in vivo.

To identify genes linked to the temperature-dependent phenotypes of *P. aeruginosa* CF39S, we carried out a comparative genomic analysis of strains CF39 and CF39S (see "Methods"). This analysis revealed only one high-confidence single nucleotide polymorphism (SNP) between the two strains. This SNP occurred in a gene encoding a putative GGDEF-domain protein that we named the thermosensory diguanylate cyclase A (*tdcA*) (Fig. 1d). *P. aeruginosa* CF39 possesses a base pair deletion in *tdcA* (162ΔG), resulting in a frameshift and truncation of TdcA. The correlation between *tdcA* and temperature-dependent phenotypes of *P. aeruginosa* CF39S was verified using a genetic linkage analysis. Two-step allelic exchange[35]—in which *tdcA*$_{162\Delta G}$ and *tdcA* were replaced with *tdcA* and *tdcA*$_{162\Delta G}$ in strains CF39 and CF39S, respectively—reversed the temperature-dependent phenotypes of these isolates (Fig. 1e). Moreover, miniTn7 transposons that were engineered bearing *tdcA* as part of its predicted polycistronic operon, or alternatively, as a fusion between the promoter of that operon and the *tdcA* open reading frame (ORF), were sufficient to confer heat-activated biofilm production to a recipient *P. aeruginosa* PAO1 laboratory strain (Fig. 1e), which naturally lacks *tdcA*. By contrast, *P. aeruginosa* PAO1 acquiring *tdcA*$_{162\Delta G}$ did not display heat-activated biofilm production (Fig. 1e). Finally, *P. aeruginosa* PAO1 engineered to express *tdcA* also exhibited thermosensitive synthesis of intracellular c-di-GMP (Supplementary Fig. 1). Taken together, these data demonstrate that *tdcA* is linked to thermal control of biofilm formation and c-di-GMP synthesis in *P. aeruginosa*.

**TdcA homologs are conserved among bacteria.** Bioinformatics indicate that *tdcA* is encoded on a Tn7-like transposon inserted in the chromosome (Fig. 1d), providing a c-di-GMP signaling module in the accessory genome that gives a gain-of-function to *P. aeruginosa* CF39S. Other genes in this transposon have identity and synteny with the locus of heat resistance (LHR), a horizontally acquired genomic island first identified in *Escherichia coli*[36], and later found in many other bacterial species[37–39]. Genes of the LHR, including orthologous genes of the *P. aeruginosa* CF39S *tdcA* operon, protect cells from heat shock[39–41], pressure treatment[41], chlorine and oxidizing chemicals[42] (Fig. 1d). A gene for a small heat shock protein (*shsp20c*) orthologous to one adjacent to the *tdcA* operon is known to confer heat tolerance to *P. aeruginosa*[43]. However, the best studied LHR from *E. coli* AW1.7 (ref. [44]) lacks an ortholog of *tdcA* (Fig. 1d).

The *tdcA* gene is present in <1% of sequenced *P. aeruginosa* genomes in the *Pseudomonas* Genome Database[45] and is absent from the well-studied laboratory strains PA14 and PAK in addition to PAO1. However, genome-sequenced *P. aeruginosa*

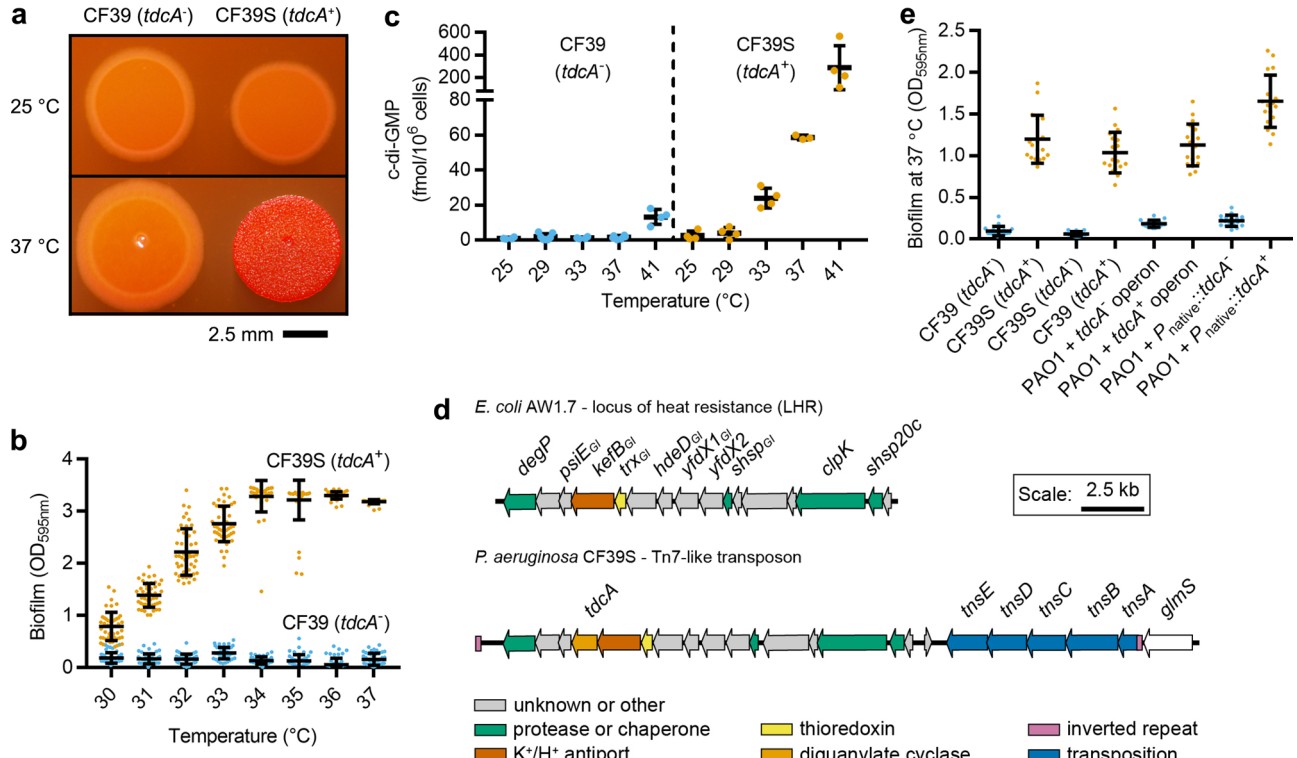

**Fig. 1 Genomics and genetic linkage analysis identify a gene for the thermosensory diguanylate cyclase (*tdcA*) in *P. aeruginosa* CF39S. a** *P. aeruginosa* CF39S displays temperature-dependent rugose colony morphology on agar. **b** *P. aeruginosa* CF39S exhibits temperature-dependent biofilm formation in microplates. Datum points represent 16 technical replicates from each of three biological replicates, and lines and bars represent means and standard deviations, respectively. **c** Liquid chromatography tandem mass spectrometry (LC–MS/MS) measurements of cellular nucleotide pools revealed that *P. aeruginosa* CF39S displays temperature-dependent increases in cyclic diguanylate (c-di-GMP). Each datum point represents an independent biological replicate, and lines and bars represent means and standard deviations, respectively, for 3–5 independent biological replicates each. **d** Schematic illustrating the Tn7-like transposon in the *P. aeruginosa* CF39S chromosome encoding *tdcA* and its synteny to the *E. coli* AW1.7 locus of heat resistance (LHR). Annotations for the *E. coli* AW1.7 genome were taken from Mercer and colleagues[44]. Open reading frames are color coded by putative functions as illustrated in the legend: gray arrows denote genes of unknown function or functions other than those illustrated here; bluish green arrows denote genes for putative proteases or chaperones; yellow arrows denote genes for thioredoxins; orange arrow denotes a diguanylate cyclase; blue arrows denote genes encoding functions for transposition; and reddish purple segments denote inverted repeats. **e** Acquisition of *tdcA* confers thermal control of *P. aeruginosa* biofilm formation in microplates. Datum points represent six technical replicates from each of three biological replicates, and lines and bars represent means and standard deviations, respectively. In **b**, **c**, and **e**, *tdcA⁻* and *tdcA⁺* strains are represented by sky blue and orange dots, respectively. Strains denoted *tdcA⁻* have the $tdcA_{162\Delta G}$ allele. $OD_{595nm}$ denotes the optical density measured at a wavelength of 595 nm.

strains encoding *tdcA* alleles have been isolated worldwide, originating from both the environment and diverse human infections (Supplementary Fig. 2 and Supplementary Data 1). BLAST searching of the National Center for Biotechnology Information (NCBI) databases using strict search and sequence analysis criteria (see "Methods") suggests that TdcA homologs are distributed among numerous environmental bacterial species as well as human and veterinary microbiota (Supplementary Fig. 3 and Supplementary Data 1). We observed that a subset of putative TdcA orthologs with >93% identity to *P. aeruginosa* CF39S TdcA ($TdcA_{Ps}$) all occur in the syntenic context of the LHR (see Supplementary Data 1 for NCBI accession numbers). By contrast, the remaining putative TdcA homologs, which share 30.2–47.2% identity with $TdcA_{Ps}$, occur in a variety of different genomic contexts. To test the bioinformatic prediction that these latter sequences encode thermosensory diguanylate cyclases, gene synthesis was used to build a construct to express a homolog identified in a subsurface metagenome[46] and belonging to a bacterium from the Betaproteobacteria order *Gallionellales* ($TdcA_{Ga}$, Supplementary Data 1, NCBI Accession No. OGS67013.1). $TdcA_{Ga}$ has 38.2% identity with $TdcA_{Ps}$. Heterologous expression of $TdcA_{Ga}$ in *P. aeruginosa* PAO1 and direct measurements of c-di-GMP in cell lysates by mass spectrometry showed that $TdcA_{Ga}$ displays temperature-dependent diguanylate cyclase activity similar to that exhibited by $TdcA_{Ps}$ (Supplementary Fig. 3). Taken together, these data suggest that TdcA orthologs and proteins having similarity to it are widespread among bacteria.

**C-di-GMP-dependent thermotransduction.** Thermotransduction is the process by which temperature is perceived by a sensory cell receptor, initiating a signaling cascade that changes cellular physiology. In *P. aeruginosa*, c-di-GMP regulates the expression of genes for synthesis of the biofilm matrix, which includes operons for extracellular polysaccharides (*pel* and *psl*) and an adhesin (*cdr*)[47,48] (Fig. 2a). We hypothesized, therefore, that c-di-GMP-dependent thermotransduction would upregulate the expression of these gene loci in strains bearing *tdcA*, driving expression of the biofilm matrix components PEL, PSL, and CdrA in response to changing temperatures. Western blotting using antisera against the proteins necessary for PEL and PSL synthesis revealed a striking upregulation of PelC, PelD, and PslG for *P. aeruginosa* CF39S when incubation temperatures were increased from 30 to 37 °C. A similar trend was observed for PAO1 engineered to express *tdcA* (Fig. 2b). *P. aeruginosa* strains

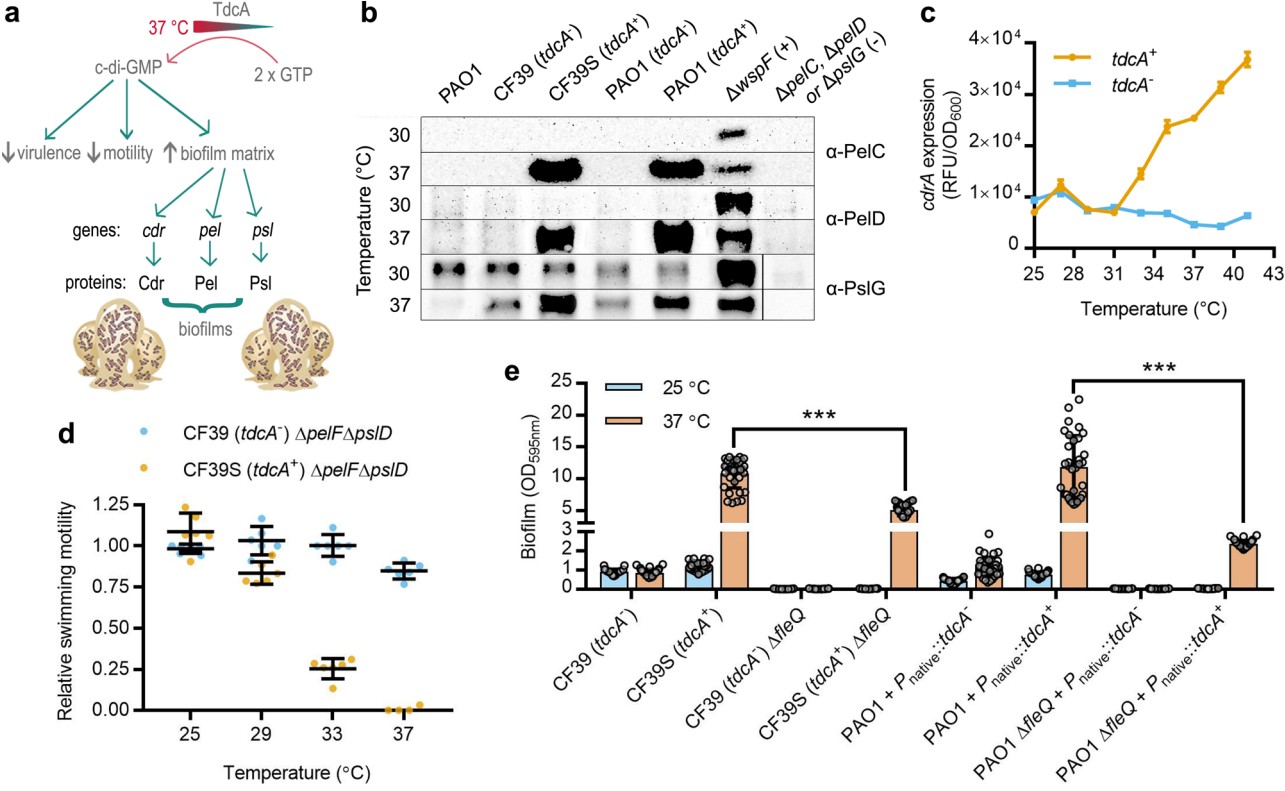

**Fig. 2 Cyclic diguanylate (c-di-GMP)-dependent thermotransduction. a** Schematic of the heat-activated c-di-GMP pathway that drives biofilm development in *P. aeruginosa* CF39S. **b** Western blots for c-di-GMP-regulated Pel and Psl proteins. Measurements were performed in biological triplicate and one representative blot is shown (original, unedited blot images are provided in Supplementary Information). Lanes were loaded with a normalized quantity of total cellular protein. The Δ*wspF* strain, which has constitutively high c-di-GMP levels[23, 32], was used as a positive control. **c** Expression of the $P_{cdrA}::gfp$ reporter. Data represent means and standard deviations of measurements each performed in biological and technical triplicate. The *tdcA*− and *tdcA*+ strains are represented by sky blue and orange lines, respectively. $OD_{600nm}$ denotes the optical density measured at a wavelength of 600 nm, and RFU denotes relative fluorescence units. **d** Thermal control of swimming motility in *P. aeruginosa* CF39S. Measurements were made in Δ*pelF* Δ*pslD* backgrounds to limit interference from extracellular polymers in motility phenotypes. Each datum point represents an independent biological replicate, and lines and bars represent means and standard deviations, respectively, for 4–6 independent biological replicates; *tdcA*− and *tdcA*+ strains are represented by sky blue and orange dots, respectively. **e** Biofilm assays indicate that TdcA integrates into the *P. aeruginosa* c-di-GMP regulatory network, in part, via the c-di-GMP-binding transcription factor, FleQ. Datum points represent 12 technical replicates from each of three biological replicates, and lines and bars represent means and standard deviations, respectively. Temperatures of 25 °C and 37 °C are represented by sky blue and orange bars, respectively. $OD_{595nm}$ denotes the optical density measured at a wavelength of 595 nm. Significant difference via two-tailed Student's *t*-test (*** denotes $P < 0.001$). Strains denoted *tdcA*− have the $tdcA_{162\Delta G}$ allele.

bearing $tdcA_{162\Delta G}$ as well as wild-type PAO1 did not display these phenotypes (Fig. 2b). By contrast, a *P. aeruginosa* PAO1 strain containing an activating mutation in the Wsp surface-sensing system (Δ*wspF*, which has constitutively elevated intracellular c-di-GMP levels (refs. [23,32])), exhibited high abundance of PelC, PelD, and PslG at all temperatures tested (Fig. 2b). Additionally, a bioreporter system, which is comprised of a plasmid-borne fusion between the *cdrA* promoter ($P_{cdrA}$) and the gene for green fluorescent protein (*gfp*)[49], indicated that *cdrA* transcription in a PAO1 strain bearing *tdcA* increased linearly 5.2-fold from 31 to 41 °C (Fig. 2c). Taken together, these data indicate that TdcA can differentially regulate a known c-di-GMP-dependent regulon in *P. aeruginosa*, and that this occurs in response to changing temperatures.

In *P. aeruginosa*, the c-di-GMP-binding transcription factor FleQ derepresses *pel*, *psl*, and *cdr* operons when c-di-GMP levels are high, and activates transcription of flagellar genes when c-di-GMP levels are low[47,48,50,51]. Unsurprisingly, we observed that strains bearing *tdcA* exhibited reduced swimming motility as temperatures were increased from 25 to 37 °C (Fig. 2d). We thus hypothesized that TdcA might integrate into the *P. aeruginosa* c-di-GMP-dependent regulon via FleQ, and reasoned that a Δ*fleQ*

mutation should diminish biofilm production by *tdcA*+ strains. Corresponding to this hypothesis, while a *P. aeruginosa* CF39S Δ*fleQ* mutant displayed temperature-dependent biofilm formation, the quantity of biofilm it produced was significantly less than it was for wild-type CF39S (Fig. 2e). Similar observations were made for a PAO1 Δ*fleQ* mutant expressing *tdcA* (Fig. 2e). Taken together, an interpretation is that TdcA partly integrates into the *P. aeruginosa* c-di-GMP regulon via FleQ, and the partial loss-of-function phenotypes of the Δ*fleQ* strains indicate that additional regulatory mechanisms—such as allosteric modulation of the Pel synthase[52,53], flagellar function[54,55], and/or unknown others—are at play.

To visualize the dynamics of c-di-GMP-dependent thermotransduction, we designed and built a heterologous system for TdcA in *Pseudomonas fluorescens*, which possesses *fleQ* but lacks *tdcA*. This organism was engineered to express *tdcA* (or the inactive $tdcA_{162\Delta G}$ allele) via site-directed transposon mutagenesis[56]. Subsequently, these strains were transformed with a luminescent c-di-GMP bioreporter, comprised of a fusion between $P_{cdrA}$ and the *luxCDABE* operon[49]. These transformants were then grown side-by-side in Petri dishes at 25 °C, then shifted to 37 °C and photographed in the dark via time-lapse imaging.

*P. fluorescens* engineered with *tdcA* and the c-di-GMP bioreporter became luminescent within 30 min of the temperature increase, whereas strains bearing $tdcA_{162\Delta G}$ or vector controls remained dark (Supplementary Movie 1). These data illustrate that TdcA-dependent thermotransduction drives physiological changes that initiate within minutes and accrue in magnitude over the course of hours.

**Thermal control of *P. aeruginosa* virulence.** Multiple mechanisms of bacterial pathogenesis are regulated by c-di-GMP[57]. Evidence suggests that high c-di-GMP levels reduce expression of the cytotoxic type III secretion system[58], produce modifications to lipopolysaccharide that affect immunogenicity[59], and decrease expression and function of the flagellum (ref. [32] and Fig. 2d), which is a key determinant of acute virulence[26,60–62]. The exopolysaccharide PSL, which is also regulated by TdcA (Fig. 2b), affects virulence[63–65] and mediates phagocytic evasion[66] and antimicrobial tolerance[67]. We hypothesized, therefore, that TdcA-dependent thermotransduction could also function to regulate *P. aeruginosa* infection outcomes. Here, we evaluated the lethality of $tdcA^+$ and $tdcA_{\Delta 162G}$ strains in the waxworm *Galleria mellonella*. A key advantage of the *G. mellonella* model is that it is possible to vary external temperatures between 25 and 37 °C without adversely affecting insect survival (Supplementary Fig. 4). We observed that experimental infections with strains bearing $tdcA^+$ and $tdcA_{162\Delta G}$ alleles produced similar mortality at 25 °C. However, *G. mellonella* infected with the *P. aeruginosa* $tdcA^+$ strain showed significantly lower mortality at 37 °C than those insects infected with a $tdcA_{162\Delta G}$ strain (Supplementary Fig. 4). Attenuation of acute virulence was similarly observed in a *Caenorhabditis elegans* infection model when inoculating bacteria were grown at 37 °C as opposed to 25 °C (Supplementary Fig. 5). Taken together, while we have not yet determined which c-di-GMP-dependent determinants affect lethality in insects and roundworms, these results demonstrate that TdcA is a temperature-dependent regulator of *P. aeruginosa* virulence.

**A thermosensory diguanylate cyclase.** So, what could account for the temperature-dependent synthesis of c-di-GMP in bacterial cells? Sequence analyses predict that TdcA is a 38.7 kDa protein of 342 amino acids that is comprised of an N-terminal Per-Arnt-Sim sensory domain (categorized as PAS3 in the Pfam database (ref. [68]), see "Discussion") connected by an α-helical linker to a GGDEF domain (Fig. 3a). The biochemistry of GGDEF domains is well documented[14–17]. These predominantly prokaryotic domains catalyze the formation of c-di-GMP from two GTP, releasing two pyrophosphate ($PP_i$) molecules as a byproduct. TdcA has a GGEEF catalytic motif. It also has a consensus I-site, which, based on prior studies of diguanylate cyclases, is known to bind to c-di-GMP and inhibit catalysis above a threshold c-di-GMP concentration[16]. By contrast, PAS sensory domains are found throughout all domains of life and are known to respond to a variety of stimuli, including oxygen ($O_2$), nitric oxide (NO), and light[69]. However, there is neither a report of a thermosensory PAS domain nor another evolutionarily conserved, putative cytosolic bacterial protein domain that functions in heat sensing.

A clue about mechanism emerged when the predicted diguanylate cyclase activity of TdcA was evaluated in vitro. Initially, TdcA was produced and purified as a recombinant fusion with hexahistidine ($His_6$) and the N-utilizing substance (NusA) solubility tag ($His_6$-NusA-TdcA). Diguanylate cyclase activity was measured using an established enzyme-coupled pyrophosphatase assay[70]. We observed that $His_6$-NusA-TdcA exhibited little, if any, activity at 22 °C over a range of GTP concentrations (Fig. 3b). Nevertheless, repeating these assays at 37 °C revealed that this

protein possessed robust diguanylate cyclase activity (Fig. 3b). TdcA also displayed an activity maximum between 0.5 and 1 mM GTP (Fig. 3b), which are GTP concentrations that approximate known intracellular GTP levels in bacteria[71]. Similar temperature-dependent activity was observed for TdcA expressed as a fusion of $His_6$ and maltose binding protein ($His_6$-MBP-TdcA) (Fig. 3c). C-terminal truncation of this protein to remove the GGDEF domain ($His_6$-MBP-TdcA$^{M167*}$) or alanine mutation of the catalytic GGEEF ($His_6$-MBP-TdcA$^{E257A}$) motif abolished its activity (Fig. 3c). Finally, UV–Vis spectrophotometry of purified TdcA did not reveal a spectral signature for a heme or flavin cofactor, which might otherwise suggest a role in $O_2$, NO, or light sensing (Supplementary Fig. 6). We postulated, therefore, that TdcA is a heat-sensing diguanylate cyclase, and moreover, that this behavior might not require a cofactor.

All enzymes display temperature-dependent catalytic rates, and most enzymes show highly similar rate–temperature dependencies[72]. A measure of these dependencies is the $Q_{10}$ temperature coefficient, which is the fold-change in the reaction rate as a consequence of increasing temperature by 10 °C. With only a few exceptions—notably including the thermosensitive transient receptor potential (thermoTRP) proteins of the peripheral nervous system in animals[19,73,74]—nearly all enzymes display a $Q_{10}$ value between two and three (Fig. 3d)[72]. Our data suggest that similar to the thermoTRPs[19–21], TdcA is an outlier to this principle (Fig. 3b). Therefore, we assessed the catalytic rate of TdcA-mediated GTP hydrolysis over a range of temperatures, and compared it to the well characterized *P. aeruginosa* diguanylate cyclase, WspR[70,75], which is involved in surface sensing.

To begin, temperature-dependent catalytic activity was measured for an equivalent number of enzyme units of WspR and TdcA using the pyrophosphatase assay. One enzyme unit was arbitrarily defined as one micromole of GTP hydrolyzed per minute at 37 °C. WspR displayed canonical temperature-dependent kinetics with a maximal $Q_{10}$ value of 2.0 between 28 and 37 °C (Fig. 3d, e). By contrast, TdcA activity was undetectable at 25 °C, yet increased with a maximal $Q_{10}$ value of 10.5 between 28 and 37 °C (Fig. 3d, e). A similar result was obtained for TdcA when c-di-GMP production was directly measured using mass spectrometry (Fig. 3f). However, repeating temperature coefficient calculations with this more sensitive method, which enabled the detection of c-di-GMP production in vitro at 25 °C (Fig. 3f), indicated that TdcA has a $Q_{10} = 135$ when calculated between 25 and 33 °C (Fig. 3d, see "Methods" and "Discussion"). These results demonstrate that TdcA is not only a bona fide diguanylate cyclase, but also highly thermosensitive.

**The thermosensitive Per-Arnt-SIM (thermoPAS) domain.** We next sought to investigate the putative role for the TdcA PAS3 domain ($PAS_{TdcA}$) in temperature sensing. Proteins containing PAS domains share commonalities that enable the engineering of artificial proteins with designer functionalities[76–78]. For instance, stimuli cause changes in the conserved PAS fold that are transmitted to the PAS-linked effector domain via torque of a linker region[69] (Figs. 3a and 4a). We hypothesized, therefore, that $PAS_{TdcA}$ could be fused to other effectors via its linker to create a chimeric protein with thermosensitive activity.

To test this hypothesis, $PAS_{TdcA}$ was fused to the GGDEF domain of WspR ($GGDEF_{WspR}$) to build a synthetic thermosensory protein (Stp1, Fig. 4a and Supplementary Methods). The activity of Stp1 was assessed in *P. aeruginosa* PAO1 cells engineered to express this protein from an arabinose-inducible expression cassette ($araC$-$P_{BAD}$). Here, expression of a constitutively actively form of WspR (WspR$^{V72D}$) (ref. [70]) from the

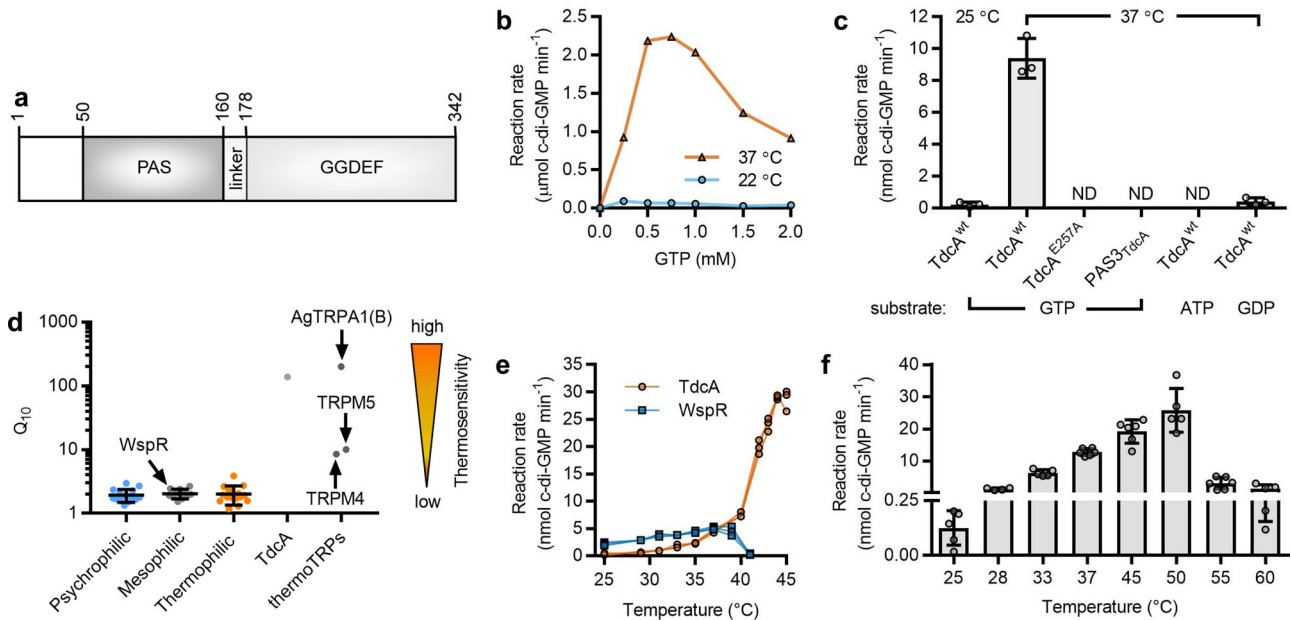

**Fig. 3 TdcA displays thermosensitive catalytic activity. a** Schematic illustrating the predicted domain organization of TdcA. **b** Substrate-dependent enzyme kinetics of His$_6$-NusA-TdcA. Measurements were performed in triplicate and one independent replicate is shown. **c** GTP is the substrate for His$_6$-MBP-TdcA. Each bar represents the mean and standard deviation of three independent replicates. ATP, adenosine-5′-triphosphate; GDP, guanosine-5′-diphosphate; GTP, guanosine-5′-triphosphate. **d** TdcA is an outlier to theory for universal enzymatic rate–temperature dependencies in bacteria. Each datum point represents a $Q_{10}$ value for an enzyme: $Q_{10}$ values for WspR and TdcA were calculated in the present work, whereas all other temperature coefficients are derived from published literature values and grouped according to the temperature-dependent activity maximum of the enzyme (psychrophilic, mesophilic, thermophilic)[72]. Lines and bars represent means and standard deviations, respectively, for 11–16 literature values in each category. **e** Temperature-dependent enzyme kinetics of His$_6$-MBP-TdcA. Measurements were performed in technical triplicate and datum points and lines represent individual replicates. **f** LC–MS/MS measurements of temperature-dependent c-di-GMP production by recombinant His$_6$-MBP-TdcA in vitro. Each datum point represents an independent replicate, and lines and bars represent means and standard deviations, respectively.

*araC-P$_{BAD}$* cassette resulted in intracellular c-di-GMP levels that were similar at both 25 °C and 37 °C (Fig. 3b). By contrast, expression of TdcA resulted in an increase in c-di-GMP levels between these two temperatures, as did expression of Stp1 (Fig. 4b). Although relative levels of c-di-GMP differed between TdcA- and Stp1-overexpressing bacteria, there was a similar 6.5- to 8.1-fold relative increase in intracellular c-di-GMP levels in both strains between 25 and 37 °C (Fig. 4b). We infer from these findings that PAS$_{TdcA}$ is a thermosensitive Per-Arnt-Sim domain (which we denote "thermoPAS," after the nomenclature of the thermoTRP), and that it transduces thermal stimuli to regulate the diguanylate cyclase activity of the adjacent GGDEF domain in TdcA.

To corroborate the observation that thermoPAS functions in temperature sensing, we coupled the N-terminus of TdcA, which encompasses the thermoPAS domain and its linker (TdcA$_{1-177}$), to *E. coli* LacZ (Fig. 4c). To do this, TdcA$_{1-177}$ was fused between the LacZ segments that otherwise correspond to the α-peptide (LacZ$_{1-92}$) and the Ω-fragment (or α-acceptor) of β-galactosidase, resulting in a temperature-sensitive enzyme that we named "thermo-gal." This chimeric enzyme displayed temperature-sensitive catalytic activity when expressed in *E. coli* LMG194, which is a strain that lacks the entire *lac* operon as well as the capacity to metabolize L-(+)-arabinose, the latter of which was used as an inducer of thermo-gal expression (Fig. 4c). By contrast, a similar fusion between the PAS2 domain of *E. coli* PdeO and LacZ resulted in a protein that retained catalytic activity when temperature was increased. PAS2 from PdeO (also called DosP) is a well-studied, heme-bound, O$_2$-sensing PAS domain[79] that is not expected to respond to temperature. SDS-PAGE analysis indicated that these fusion proteins are not expressed in a temperature-dependent fashion in *E. coli* (Supplementary Fig. 7),

suggesting that differences in activity do not result from changes in protein abundance. Taken together, these data provide evidence that thermoPAS constitutes a molecular thermosensory device that is linked to the temperature-sensing functionality of TdcA.

## Discussion

Enzymes exhibiting thermosensitive reaction rates that are regulated by a thermoPAS domain exemplify a previously unreported mechanism for cellular temperature sensing. It is possible that the thermoPAS domain may be present in numerous and diverse proteins. PAS domains are characterized by an evolutionarily conserved three-dimensional fold yet share little primary sequence identity[80]. Perhaps the most valuable tool that might predict which proteins contain a thermoPAS domain is the Pfam database[68], which uses hidden Markov models to define protein domains. The Pfam PAS domain clan contains 17 member families (https://pfam.xfam.org/clan/PAS_Fold). The thermoPAS domain belongs to the Pfam PAS3 family (https://pfam.xfam.org/family/PF08447) (Supplementary Fig. 3). As of October 14, 2020, the Pfam database[68] lists 45,587 sequences that can be grouped into 8526 protein architectures that contain PAS3. Although PAS3 is found in proteins from all kingdoms of life, in bacterial proteins PAS3 can be found linked to GGDEF and EAL domains (that are part of diverse multidomain proteins), methylating chemotaxis protein signal (MCPsignal) domains, histidine kinase (HisKA and HTPase_C) domains, and stage II sporulation protein E (SpoIIE) domains, among others. Even if only a small portion of these PAS3 domains function in temperature sensing, then it is possible that thermoPAS might regulate not only c-di-GMP synthesis and degradation, but also bacterial taxis, two-component signal transduction, and sporulation.

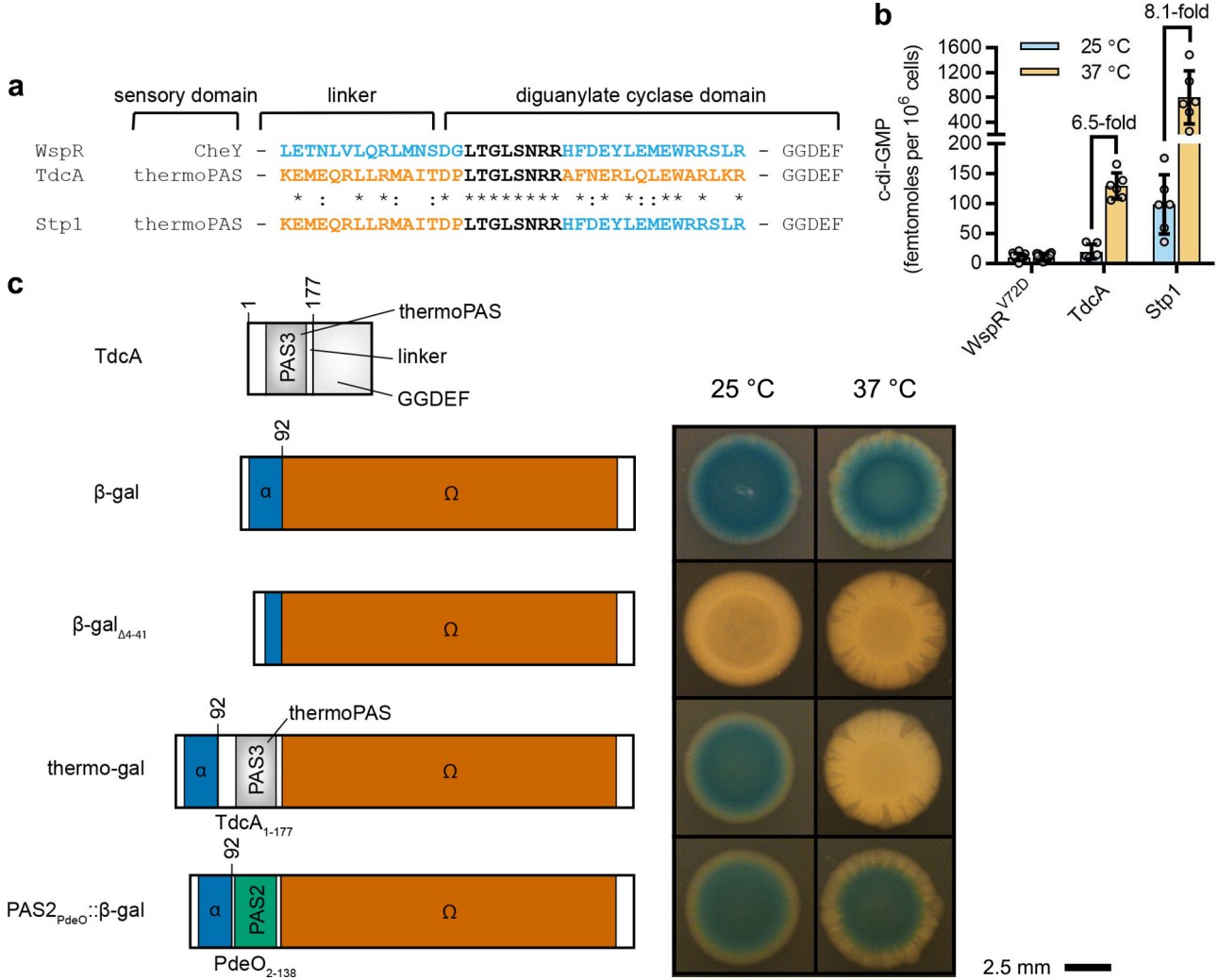

**Fig. 4 Engineered protein chimeras identify a thermosensitive Per-Arnt-SIM (thermoPAS) domain. a** CLUSTALW sequence alignment of WspR (blue) and TdcA (orange) protein sequences. The TdcA thermoPAS domain was fused to the GGDEF domain of WspR via the region of identity at the N-terminus of the GGDEF domains (black), yielding the chimeric synthetic thermosensory protein 1 (Stp1). **b** LC–MS/MS measurements of c-di-GMP levels in cells expressing WspR, TdcA, and Stp1 from an arabinose-inducible expression cassette ($araC$-$P_{BAD}$). Each datum point represents an independent biological replicate, and lines and bars represent means and standard deviations, respectively, for 6–12 independent biological replicates. Temperatures of 25 °C and 37 °C are represented by sky blue and orange bars, respectively. **c** Internal fusions of the N-terminal portion of TdcA to LacZ (β-galactosidase) produce an enzyme, thermo-gal, which displays temperature-sensitive catalytic activity when expressed in *E. coli* LMG194. Bacteria were grown on LB agar containing X-gal and 1% L-(+)-arabinose.

TdcA kinetics challenge theory for universal enzymatic rate–temperature dependencies[72], yet outliers to this theory are not without precedent or biological relevance. One example includes the thermoTRPs, which couple changes in temperature to ion channel gating to enable the sensation of hot and cold temperatures by sensory neurons[20,21]. While it is still not clear what part of thermoTRPs sense temperature[81,82], the $Q_{10}$ temperature coefficient has been used to describe the fold-change in electrical current conducted by these proteins per 10 °C change[20,81,83]. Various thermoTRP isoforms exhibit diverse $Q_{10}$ values, ranging from ~4 to >200 (ref. [19]), which bound those of TdcA ($Q_{10} = 135$ in the temperature range of 25–33 °C). Because $Q_{10}$ values depend on the temperature range used to make the calculation[72], we acknowledge that this may be a conservative estimate of a physiologically relevant $Q_{10}$ value for TdcA, which is >2400 in vitro when the calculation is repeated using liquid chromatography tandem mass spectrometry (LC–MS/MS) measurements (Fig. 3f) at 25 and 28 °C (see "Methods"). Using $Q_{10}$ values as a gauge for thermosensitive

biomolecular behavior, we propose, therefore, that thermosensitive catalysis of a second messenger constitutes a rudimentary biochemical mechanism for thermal sensation in bacteria, and furthermore, that c-di-GMP-dependent thermotransduction enables bacteria to regulate diverse aspects of their physiology and social behaviors in response to temperature.

## Methods

**Strains, physiological buffers, and microbiological media**. Bacterial strains used in this study are listed in Supplementary Table 1. *P. aeruginosa*, *P. fluorescens*, and *E. coli* were routinely grown in lysogeny broth (LB)[84], which contained, per liter of milliQ water, 10.0 g tryptone, 5.0 g yeast extract, and 5.0 g NaCl. No-salt lysogeny broth (NSLB) contained, per liter of milliQ water, 10.0 g tryptone and 5.0 g yeast extract, and was supplemented with 15% w/v sucrose when required. Vogel–Bonner Minimal Medium (VBMM)[85] was prepared as a 10× concentrate, which contained per liter of milliQ water 2.0 g MgSO$_4$·7H$_2$O, 20 g citric acid, 100 g K$_2$HPO$_4$, and 35 g NaNH$_4$H-PO$_4$·4H$_2$O, and was adjusted to pH 7.0 and sterilized by filtration. The 10× VBMM solution was diluted as required in sterile milliQ water. A 1 ml aliquot of a 1000× trace metals solution was added per liter of VBMM. The 1000× trace metals solution contained 2 mM FeCl$_3$·6H$_2$O, 1 mM MnCl$_2$·4H$_2$O, 0.05 mM Zn(NO$_3$)$_2$·6H$_2$O, 1 mM CaCl$_2$, 0.1 mM CoSO$_4$·7H$_2$O, 0.1 mM CuCl$_2$·2H$_2$O, 0.1 mM Na$_2$MoO$_4$·2H$_2$O, 2 mM

MgSO$_4$·7H$_2$O, and 0.2 mM NiCl$_2$. Semisolid media were prepared by adding 1.5% agar to LB, 1.0% noble agar to VBMM, respectively. Tryptone agar contained, per liter of milliQ water, 10.0 g tryptone and 1.0% agar. Superbroth[86] contained, per liter of milliQ water, 32.0 g tryptone, 20 g yeast extract, 5 g NaCl, and 5 ml 1 N NaOH. Phosphate-buffered saline (PBS) was purchased as a 20× concentrate (Amresco) and was diluted as needed in sterile milliQ water. Unless noted otherwise, all media, buffers, and water were sterilized using an autoclave by standard operating procedures.

Antibiotics were added as required at the following activity-corrected concentrations: for *E. coli*, gentamicin (Gm) at 10 μg/ml, carbenicillin (Cb) at 50 μg/ml, tetracycline (Tc) at 10 μg/ml, and kanamycin (Km) at 50 μg/ml; for *P. aeruginosa* PAO1: Cb at 300 μg/ml; Gm at 60 μg/ml for merodiploid and plasmid selection, or Gm at 30 μg/ml for miniTn7 mutants and Tc at 100 μg/ml for miniCTX mutants; for *P. aeruginosa* CF39 and CF39S: Gm at 1000 μg/ml. Where indicated, Congo red was added to media at 40 μg/ml; for *P. fluorescens* Pf-81: Gm at 30 μg/ml for miniTn7 mutants and Km at 500 μg/ml.

**Photography of bacterial colonies on agar plates**. Overnight cultures of *P. aeruginosa* were grown in LB at 25 °C and 250 r.p.m. These cultures were diluted 1:1000 in LB, and 3 μl of this inoculum was aliquoted onto tryptone agar containing Congo red. These Petri dishes were wrapped in Parafilm® and placed in an incubator set to either 25 °C or 37 °C and incubated for 18 h. *E. coli* LMG194 strains expressing LacZ fusions (pTER132-1, pTER145-1, pEA04-1, or pTER143-1, Supplementary Table 2) were grown on LB + Cb agar containing 158 μM 5-bromo-4-chloro-3-indolyl-β-D-galactopyranoside (X-gal) and 1% w/v L-arabinose. These agar plates were inoculated with a 3 μl aliquot of *E. coli* LMG194 cells that had been collected from LB + Cb agar and suspended in PBS at an optical density at 600 nm (OD$_{600}$) of 0.3. The plates were then grown at 25 °C or 37 °C until sufficient color had developed in the colonies (~2–3 days). Images of surface-illuminated colonies were captured using a digital camera mounted on a Nikon® AZ100 dissection microscope using a 0.5× objective and NIS Elements® v.4.13.00. Images were enhanced for contrast and brightness using Photoshop CS3 (Adobe).

Luminescent *P. fluorescens* strains were streaked onto LB agar and grown at 25 °C overnight. Time-lapse images of bioluminescence were captured using a FluorchemQ gel documentation system with Alphaview (v3.4.0) software (Proteinsimple®). Images were assembled into animations using iMovie v10.2.2.

**Standard molecular and biochemical methods**. All plasmids and primers used in genetic manipulations are listed in Supplementary Tables 2 and 3, respectively. All basic microbiological and molecular procedures were executed according to standard protocols[87]. Genomic DNA (gDNA) isolation, plasmid preparation, and DNA gel extraction were performed using nucleotide purification kits purchased from Qiagen or BioBasics. All restriction enzymes, T4 DNA ligase, Quick ligase, Taq DNA polymerase, RNase A, Antarctic phosphatase, and shrimp alkaline phosphatase were purchased from New England Biolabs. Phusion DNA polymerase was purchased from ThermoFisher Scientific. Transformations of *P. aeruginosa* and *P. fluorescens* were carried out using established protocols for electroporation[88]. Site-directed mutagenesis of plasmids was carried out using the QuikChange II XL Site-Directed Mutagenesis Kit (Agilent). Oligonucleotide primers were purchased from Integrated DNA Technologies. Protein concentrations were measured using the Pierce 660 nm protein assay with the addition of the ionic detergent compatibility reagent (IDCR) when necessary (ThermoFisher Scientific).

**Swimming motility assays**. Motility assays were performed using LB containing 0.3% agar. Each Petri dish containing this medium was "stab" inoculated with four strains: TER43-4G (CF39 *ΔpelF ΔpslD*), TER44-6A (CF39S *ΔpelF ΔpslD*), JJH0 (wild-type PAO1), and JJH283 (PAO1 *ΔfliC*) (Supplementary Table 1). Stab inoculation was executed using sterile toothpicks to transfer a colony of each strain that had been freshly streaked on LB agar into the LB containing 0.3% agar. These plates were grown at 25, 29, 33, or 37 °C for 18 h and the diameter of motility zones were measured using a ruler. To correct for differences in growth rates at the different temperatures, relative motility (mot$_R$) was calculated using the following equation:

$$\text{mot}_R = \frac{\left(\text{mot}_{\text{test}} - \text{mot}_{\Delta fliC}\right)}{\left(\text{mot}_{\text{wt}} - \text{mot}_{\Delta fliC}\right)} \quad (1)$$

where mot$_{\text{test}}$, mot$_{\text{wt}}$, and mot$_{\Delta fliC}$ are the motility zone diameters of the test strains (CF39 or CF39S), wild-type PAO1 (positive control), and the PAO1 *ΔfliC* mutant (negative control), respectively.

**Microplate biofilm formation assays**. Microplate biofilm formation assays were executed using established methods[89,90]. Here, *P. aeruginosa* was streaked out from a cryogenic stock onto LB agar and grown at 30 °C overnight. The next day, bacterial cells were collected from the LB agar with a sterile cotton swab and suspended in LB to an optical density at 600 nm (OD$_{600}$) of 0.3. This suspension was then diluted 1:30 in LB, and 100 μl aliquots were transferred to the wells of a 96-well microplate. The outer wells of the microplate were filled with 100 μl of milliQ water. These microplates were incubated in a humidified chamber for 8 h. Following incubation, microplate wells were washed three times with milliQ water

and air-dried for 10 min. Subsequently, 110 μl of 0.1% crystal violet solution (containing 0.1% w/v crystal violet, 6.3% v/v ethanol, 0.16% v/v phenol, and 0.3% v/v methanol) was added to each well. Biomass was stained for 10 min at room temperature and then the crystal violet solution was removed using a multichannel micropipette. Microplates were then washed twice with milliQ water and left to air dry overnight. Biomass-bound crystal violet was solubilized by adding 110 μl of 30% acetic acid to each well. Lastly, 100 μl aliquots of the solubilized crystal violet solution were transferred to a new 96-well microplate. Optical density at 550 nm (OD$_{550}$) or 595 nm (OD$_{595}$) was measured using an Enspire® microplate reader (Perkin-Elmer).

**Construction of allelic exchange vectors for antibiotic-resistant *P. aeruginosa* isolates**. Initially, allelic exchange vectors for *tdcA* and *tdcA*$_{162\Delta G}$ were constructed in pEX18Ap (Supplementary Table 2) using established protocols[91]. Here, primers oLS14 and oLS15, which were tailed with BamHI and PstI restriction sites, respectively, were used to clone *tdcA* and *tdcA*$_{162\Delta G}$ from CF39S and CF39 gDNA, respectively. The PCR products were gel purified, digested with BamHI and PstI, and then ligated into pEX18Ap that had been similarly digested with restriction enzymes. The ligation products were transformed into *E. coli* DH5α using electroporation, and transformants were selected on LB + Cb agar. An established protocol for colony PCR of pEX18Ap using M13F and M13R primers[35] was used to identify clones with the desired inserts, yielding pLS6 and pLS11, which contained the *tdcA*$_{162\Delta G}$ and *tdcA* alleles, respectively (Supplementary Table 2). The sequence of the inserts in pLS6 and pLS11 was verified by Sanger sequencing using primers M13F and M13R (Supplementary Table 2).

To facilitate simpler genome editing of antibiotic-resistant isolates (such as *P. aeruginosa* CF39 and CF39S), we created allelic exchange vectors with both an antibiotic resistance cassette and a gene encoding a fluorescent protein. This dual selection tactic enabled recovery of merodiploid colonies from agar plates by virtue of enrichment via increased antibiotic resistance and selection for colonies expressing the fluorescent protein. Dual-marker vectors were built using two different approaches.

In a first approach, the allelic exchange vector for *ΔpelF* allele was built by fusing a constitutively expressed gene for mCherry onto the 3′-end of a *ΔpelF* deletion allele. To begin, regions upstream and downstream of *P. aeruginosa* CF39S *pelF* were cloned using primer pairs oJJH1435/oJJH1436 and oJJH1437/oTER232 (Supplementary Table 3), respectively. A third fragment containing the P$_{01/03/04}$-RBSII-mCherry promoter–gene fusion was cloned from pBT277 (Supplementary Table 2) using primers oTER233/oTER234 (Supplementary Table 3). Primers oJJH1436 and oTER232 were synthesized with extensions that were homologous to the 5′-ends of the adjoining fragments. Primers oJJH1435 and oTER234 were tailed with *attB1* and *attB2* sequences, respectively, for Gateway® cloning (Supplementary Table 3). The three PCR fragments were gel purified and then joined using splicing-by-overlap-extension (SOE) polymerase chain reaction (PCR)[92] using primer pair oJJH1435/oTER234. This PCR product was gel purified and then recombined with pDONRPEX18Gm using BP Clonase II (Invitrogen) according to the manufacturer's directions. The recombination products were transformed into *E. coli* DH5α using electroporation, and transformants were selected on LB + Gm agar. Lastly, an established protocol for colony PCR[35] of pDONRPEX18Gm using M13F and M13R primers was used to identify a clone with the desired insert, yielding pTER124D (Supplementary Table 2). The sequence of the insert in pTER124D was verified by Sanger sequencing using primers M13F, M13R, and oJJH1437.

In a second approach, we incorporated the P$_{01/03/04}$-RBSII-GFPmut3* promoter–gene fusion into the pDONRPEX18Gm backbone, yielding pDONRPEX18Gm-GFPmut3* (Supplementary Table 2). This produced a dual-marker allelic exchange vector in which deletion alleles could be assembled by SOE-PCR from only two PCR fragments with homology to the target gene locus in the *P. aeruginosa* chromosome. To begin, the P$_{01/03/04}$-RBSII-GFPmut3* promoter–gene fusion was cloned from pBT270 (Supplementary Table 2) using primers oTER230 and oTER231 (which were tailed with a SacI restriction site). This PCR product was gel purified and digested with SacI, and subsequently, it was inserted into SacI-digested, Antarctic phosphatase-treated pDONRPEX18Gm using Quick Ligase. The ligation mixture was then transformed into *E. coli* ccdB Survival II® (Invitrogen). A colony that was brightly fluorescent under long-wavelength ultraviolet illumination was then selected and streaked out on LB + Gm + Cm agar. Plasmid DNA was purified from this culture and sequence verified using primers oTER005, oTER240, and oTER241, yielding pDONRPEX18Gm-GFPmut3*.

A *ΔpslD* allele was built by cloning the regions upstream and downstream of *P. aeruginosa* CF39S *pslD* using primer pairs oTER237/oTER236 and oTER235/oTER238. Primers oTER237 and oTER238 were tailed with *attB1* and *attB2* sequences, respectively (Supplementary Table 3). These fragments were then gel purified and fused via established protocols[35] for SOE-PCR using primer pair oTER237/oTER238. The recombination products were then inserted into pDONRPEX18Gm-GFPmut3* using BP Clonase II, and this reaction mixture was transformed into *E. coli* DH5α using electroporation. The desired construct was selected on LB + Gm agar, identified via colony PCR, and sequence verified using M13F and M13R primers as described above, yielding pTER125A (Supplementary Table 2).

A similar tactic was used to build Δ*fleQ* allele. Regions upstream and downstream of *P. aeruginosa* CF39S *fleQ* were cloned using the primer pairs oTER437/oTER438 and pTER439/oTER440. Primers oTER437 and oTER440 were tailed with *attB1* and *attB2* sequences, respectively (Supplementary Table 3). These fragments were then gel purified and fused via SOE-PCR using primer pair oTER437/oTER440. The product was then inserted into pDONRPEX18Gm-GFPmut3* using BP Clonase II, and the reaction mixture was transformed into *E. coli* DH5α using electroporation. The desired construct was then selected on LB + Gm agar, identified by colony PCR, and sequence verified using the M13F and M13R primers described above, yielding the plasmid pTER150E (Supplementary Table 2).

**Allelic exchange in *P. aeruginosa* CF39 and CF39S**. Allelic exchange for *pelF*, *pslD*, *fleQ*, and *tdcA* was conducted according to the method of Hmelo and colleagues[35] with some minor modifications for *P. aeruginosa* CF39 and CF39S. All suicide vectors were introduced into CF39 or CF39S using biparental mating with *E. coli* S.17.1 (λpir). *P. aeruginosa* CF39 and CF39S pLS6- and pLS11-derived merodiploids were selected on VBMM containing 1500 μg/ml Cb. *P. aeruginosa* pTER124D-, pTER150E-, and pTER125-derived merodiploids were enriched on VBMM containing 1000 μg/ml Gm, and subsequently selected based on expression of GFPmut3* or mCherry using a Nikon AZ100 stereomicroscope with LED light source and GFP and RFP filters, respectively. All pEX18-derived allelic exchange vectors used in this study encode *sacB*, and therefore, double-crossover mutations were selected on NSLB + 15% w/v sucrose using established procedures[35]. Mutation of a single base pair in the *tdcA* alleles of strains derived from CF39 and CF39S were identified by sequencing PCR products amplified from the *P. aeruginosa* chromosome using an established colony PCR protocol[35,93] with primer pair oLS14/oLS15. Unmarked deletion mutations of *pelF* and *pslD* were detected by gel electrophoretic analysis of PCR products from an established colony PCR protocol[35,93] with primer pairs oKMC181/oKMC182 and oKMC179/oKMC180, respectively. Unmarked deletion mutations in *fleQ* were identified via colony PCR using primer pair oTER464/oTER465. All these PCR products were gel purified and sequenced using the same primer pairs to confirm a precise deletion mutation in the *pelF*, *pslD*, or *fleQ* ORFs.

**Design and construction of a chimeric thermosensitive diguanylate cyclase**. A CLUSTAL Omega[94] alignment and PSIPRED[95,96] analysis of TdcA and WspR primary amino acid sequences revealed a conserved LTGLSNRR sequence adjacent to the linker regions between the GGDEF domains of both enzymes and their cognate sensory domains. We designed a chimeric thermosensory enzyme, therefore, by fusing the DNA sequence encoding the TdcA-PAS3-linker-LTGLSNRR region to the DNA sequence encoding the WspR GGDEF (*wspR*GGDEF) domain, the latter of which was truncated immediately adjacent to and excluded the consensus LTGLSNRR sequence. To begin, *wspR*GGDEF was cloned from *P. aeruginosa* PAO1 gDNA using primer pair oHA11/oHA12. At the same time, the nucleic acid sequence encoding the TdcA thermosensory PAS3 domain (*tdcA*PAS3) and its adjacent linker and LTGLSNRR sequence was cloned from *P. aeruginosa* CF39S gDNA using primer pair oHA13/oHA14. The PCR products were then gel purified and fused via SOE-PCR using primer pair oHA12/oHA13. The oHA12 and oHA13 primers were tailed with Gateway®-compatible *attB2* and *attB1*-sites, respectively, and thus the SOE-PCR product was gel purified and recombined with pDONR-PEX18Gm using BP Clonase II. The reaction mixture was then transformed into *E. coli* DH5α, and transformants were selected on LB + Gm agar. A clone with the correct insert was identified by colony PCR and sequenced using M13F and M13R primers, yielding the entry vector pHA110, which encoded a fusion between TdcA_PAS and WspR_GGDEF that we denoted synthetic thermosensory protein 1 (Stp1, Supplementary Table 2).

**Design and construction of thermo-gal and other chimeric β-galactosidase enzymes**. A vector for expression of *E. coli* β-galactosidase (LacZ) with a C-terminal His_6 tag was assembled in vitro from three DNA fragments. The first fragment was built in vitro by annealing the DNA ultramers oTER287 and oTER288 by adding equimolar portions of each one in Tris-EDTA (0.5 mM, pH 8.0), boiling them for 5 min, and dropping the temperature of by 1 °C every minute for 70 min. Concomitantly, primer pairs oTER285/oTER289 and oTER286/oTER290, which contain homology extensions (Supplementary Table 3), were used to amplify the pAraCpBAD/His/lacZ backbone (Supplementary Table 2) in two fragments. These three fragments were then assembled via Gibson assembly to obtain pTER132 (Supplementary Table 2). This construct was sequenced using the primers oTER277, oTER278, oTER279, oTER280, oTER281, oTER282, oTER283, oTER284, oTER306, and oTER307 (Supplementary Table 3).

A construct for expressing thermo-gal was built using Gibson assembly[97] of two PCR products. First, a PCR product encompassing the plasmid backbone and *lacZ* gene, the latter of which was bifurcated at the nucleotide corresponding to the 93rd amino acid of β-galactosidase, was amplified from pTER132 using primer pair oTER323/oTER324. Second, the DNA corresponding to TdcA_{1-177} was cloned from pTER9 using the primer pair oTER322/oTER325. Primers were tailed with homology extensions to the other fragment (Supplementary Table 3). These fragments were then joined via Gibson assembly[97], transformed into *E. coli* DH5α,

and selected on LB + Cb, yielding pTER143 (Supplementary Table 2). This plasmid was then verified by Sanger sequencing using the primers oTER057 and oTER277.

Constructs for expressing PAS2_{PdeO}-gal and β-gal^{Δ4-41} were built in an analogous fashion to pTER143 from two PCR products. For PAS2_{PdeO}-gal, the PAS2_{PdeO} fragment was cloned from *E. coli* DH5α gDNA using primer set oTER413/oTER414, and the plasmid backbone and lacZ gene were cloned from pTER132 using primer set oTER411/oTER412. These fragments were joined via Gibson assembly[97], transformed into *E. coli* DH5α, and selected on LB + Cb, yielding pEA03 (Supplementary Table 2). This plasmid was then verified by Sanger sequencing using the primers oTER057 and oTER277. A construct for β-gal^{Δ4-41} was built from two PCR products that excluded the nucleotides encoding amino acids 4 to 41 of LacZ from pTER132. These fragments were cloned using primers oTER290/oTER329 and oTER305/oTER328, which were then joined via Gibson assembly[97]. Plasmids were transformed into *E. coli* DH5α and selected on LB + Cb, yielding pTER145 (Supplementary Table 2), which was then sequenced using primers oTER277 and oTER279.

**Construction of miniTn7 vectors**. The putative 7-gene operon containing the *tdcA* allele was cloned from *P. aeruginosa* CF39 and CF39S gDNA using primer pair oJJH1213/oJJH1218 and Phusion® High-Fidelity (HF) polymerase. Primers oJJH1213 and oJJH1218 were tailed with *attB1* and *attB2* sequences, respectively. The PCR products from CF39 and CF39S gDNA were each gel purified and recombined independently with pDONR221 using BP Clonase II, yielding pKL2 (with *tdcA*_{162ΔG}) and pJJH293 (with *tdcA*), respectively. These constructs were produced by transforming the PCR products into *E. coli* DH5α, selecting transformants on LB + Kn, and identifying clones with the desired insert via colony PCR with M13F and M13R primers. The vectors pKL2 and pJJH293 were size-verified using restriction digestion, and partially sequence-verified using M13F and M13R primers. Finally, pKL2 and pJJH293 were each recombined with pUC18-miniTn7T2.1-Gm-GW (ref. [98]) using LR Clonase II (Invitrogen) according to the manufacturer's directions, yielding pKL3 and pHA215, respectively (Supplementary Table 2). Inserts in pKL3 and pHA215 were size-verified by PCR and partially sequenced using primer pair oJJH1695/oJJH1696 (Supplementary Table 3).

A fusion between the native promoter of the putative 7-gene operon containing *tdcA* (P_native) and the *tdcA* ORF was also built. Here, a 143-bp fragment encompassing the 56 bp of the putative promoter and the first 87 bp of the first gene in the 7-gene operon was cloned using primer pair oTER1/oTER2 (Supplementary Table 3). At the same time, the *tdcA* and *tdcA*_{162ΔG} ORFs along with the 40 bp preceding them (from strains CF39 and CF39S, respectively) were cloned with primer pair oTER3/oTER11 (Supplementary Table 3) and gel purified. These two PCR products were then fused independently by SOE-PCR to the fragment encoding P_native using primer pair oTER1/oTER11. The SOE-PCR products were then gel purified and recombined with pDONR221 using BP Clonase II. Clones were recovered on LB + Kn and the plasmids with the desired inserts were identified by colony PCR and Sanger sequencing using M13F and M13R primers (Supplementary Table 3). This yielded pKL1 and pTER1 (Supplementary Table 2), which contained the P_native::*tdcA*_{162ΔG} and P_native::*tdcA* from *P. aeruginosa* CF39 and CF39S, respectively. Finally, pKL1 and pTER1 were each recombined with pUC18-miniTn7T2.1-Gm-GW using LR Clonase II, yielding pKL4 and pTER9, respectively (Supplementary Table 2). Inserts in pKL4 and pTER9 were size-verified by PCR and sequenced using primer pair oJJH1695/oJJH1696 (Supplementary Table 3).

L-(+)-arabinose-inducible expression constructs for the diguanylate cyclases TdcA, WspR, and Stp1 were also built using a miniTn7 vector. Initially, primer pair oHA229/oHA230 was used to clone the *E. coli* araC-P_{BAD} controlled expression cassette from pJN105 (Supplementary Table 2). In tandem, primer pair oHA231/oHA232 was used to clone the gene *wspR*^{V72D} from pETduet::wspR^{V72D}/PA2133 (Supplementary Table 2), which encodes a constitutively active form of WspR that does not exhibit concentration-dependent activity[70]. The *tdcA* allele was cloned using oHA233/oHA234 from *P. aeruginosa* CF39S gDNA, and primer pair oHA235/oHA236 was used to clone *stp1* from the entry vector pHA110 (Supplementary Table 2). Subsequently, the vector pUC18T-miniTn7T-Gm (Supplementary Table 2) was linearized using HindIII-HF and KpnI, and the 5′ overhang was dephosphorylated using shrimp alkaline phosphatase. Alleles encoding *wspR*^{V72D}, *tdcA*, and *stp1* were independently fused with the araC-P_{BAD} controlled expression cassette and linearized pUC18T-miniTn7T-Gm using Gibson assembly[97]. Gibson cloning was facilitated by tailing primers oHA229, oHA232, oHA234, and oHA236 with homology extensions to the pUC18T-miniTn7T-Gm multiple cloning site, whereas oHA231, oHA233, and oHA235 were tailed with homology extensions to araC-P_{BAD} (Supplementary Table 3). Assembly reactions were transformed into *E. coli* DH5α and clones with the correct insert were identified by colony PCR and sequenced using primers oHA229, oHA230, oHA237, and oHA238 (Supplementary Table 3). This yielded pHA211, pHA208, and pHA210 (Supplementary Table 2), which corresponded to pUC18T-miniTn7T-Gm with araC-P_{BAD}::*wspR*^{V72D}, araC-P_{BAD}::*tdcA*, and araC-P_{BAD}::*stp1* inserted at the multiple cloning site, respectively.

An L-(+)-arabinose-inducible expression construct for the TdcA_{Ga} homolog was similarly built in a miniTn7 vector. To begin, the gene encoding TdcA_{Ga} was synthesized (Supplementary Data 1) and cloned in pDONR221 by a commercial provider (Invitrogen), yielding pDONR221::*tdcA*_{Ga} (Supplementary Table 2). The

$tdcA_{Ga}$ gene was then amplified from this plasmid using primer pair oHA274/oHA275. At the same time, primer pair oHA229/oHA230 was used to clone the *E. coli araC-$P_{BAD}$* controlled expression cassette from pJN105, as described above. Subsequently, pDONRPUC18-miniTn7T2.1-Gm (Supplementary Table 2) was digested with HindIII and KpnI-HF to linearize the plasmid. The two PCR products and plasmid backbone were then joined via Gibson assembly, and transformed into *E. coli* DH5α. Clones with the correct insert were identified by colony PCR and sequenced using primers oHA229, oHA230, oHA237, and oHA238 (Supplementary Table 3), yielding pHA245 (Supplementary Table 2).

**Site-specific miniTn7 mutagenesis of *P. aeruginosa* and *P. fluorescens*.** Transposon mutants of *P. aeruginosa* and *P. fluorescens* strains were generated through incorporation of miniTn7 vectors at the neutral *att*Tn7 site on the chromosome via co-electroporation of strains with the miniTn7 vector and helper plasmid pTNS2 (Supplementary Table 2) via standard molecular methods[99]. All miniTn7 transposon mutants engineered in this fashion are listed in Supplementary Table 1 and were selected on LB + Gm agar. The gentamicin resistance cassette (*aac1*) encoded on miniTn7 was removed from the chromosome of bacterial strains that were used for gene expression measurements. The excision of *aac1* was executed via FLP-mediated recombination with the plasmid pFLP2 (ref. [99]). Briefly, *P. aeruginosa* mutants with the chromosomally integrated miniTn7 element were electroporated with pFLP2 and transformants were selected on LB + Cb agar and incubated at 37 °C overnight. Subsequently, colonies were streaked out on NSLB agar containing 15% w/v sucrose and incubated at 30 °C overnight to select for plasmid-cured cells. Loss of *aac1* and pFLP2 was established by confirming sensitivity of the miniT7 mutant to antibiotics on LB + Gm and LB + Cb agar.

**Bioreporter measurements.** Strains transformed with the c-di-GMP bioreporter pCdrA::gfp$^S$ or the promoterless control vector pMH487 (Supplementary Table 2) were streaked onto LB + Gm agar and incubated at 25 °C overnight. A single colony from each plate was used to inoculate 5 ml of VBMM + Gm broth. These cultures were grown overnight at 25 °C at 250 r.p.m. The following day, the cultures were standardized to an $OD_{600}$ of 0.05 in 5 ml VBMM + Gm broth. Cultures were then grown at the indicated temperature (27–41 °C) in a shaking incubator at 250 r.p.m. until an $OD_{600}$ of 0.8 had been reached. Subsequently, 200 µl aliquots of these cultures were transferred into black, clear-bottom 96-well microplates. $OD_{600}$ and fluorescence (excitation at 488 nm, emission at 509 nm) measurements were taken using an Enspire® microplate reader (Perkin-Elmer). Gene expression was calculated as relative fluorescence units (RFUs), which is the total number of photon counts divided by the $OD_{600}$ measurement of the strain bearing pCdrA::gfp$^S$, background corrected by subtracting the total number of photon counts divided by the $OD_{600}$ measurement for the same strain bearing pMH487.

**Genome sequencing, assembly, and comparative genomics.** Complete genome sequencing of *P. aeruginosa* CF39 and CF39S was performed using Roche 454® instrumentation at the University of Washington via established methodology for library preparation and device operation[100]. Assembly was performed using Newbler[100] version 2.0.00.19 (454 Life Sciences, Roche Diagnostics Corporation). This approach generated 72 contigs with length greater than 500 bp and 99.72% of the bases having quality scores of 40 and above.

An in-house computational protocol for comparative genomic analysis[101] was used to identify the high-confidence differences between strain CF39 and CF39S. Briefly, Roche 454® sequencing reads from each genome were mapped to the assembled contigs of the other genome to identify regions in which multiple reads from one genome differed in sequence from the consensus sequence of the other genome. Since the consensus sequence will have a small number of errors due to both assembly errors and low coverage in some regions, high-confidence SNPs were sought that were found at the same position in the reciprocal mapping. This protocol essentially enables a search for differences between two genomes that are evidenced by multiple raw sequencing reads in both genomes. These results were filtered to retain only high-confidence differences that were supported by at least 10 reads from each genome, where at least 80% of the reads spanning that region showed the same difference[101].

Genome finishing of strain CF39S was carried out using fee-for-service library construction and PacBio® Sequencing Services at the University of Washington. Here, the CF39S genome was sequenced to ~140-fold coverage with four Single Molecule Real-Time (SMRT) sequencing flow cells using libraries built with a 3–20 kb insert size. Using 25% of the longest reads, the CF39S genome was assembled using the Hierarchal Genome Assembly Process (HGAP)[102] implemented within SMRT Analysis for PacBio® RSII v.2.3.0. The finished genome assembly of strain CF39S was verified by generating a BamHI-restricted optical map, which was carried out on a fee-for-service basis by OpGen® (Gaithersburg, Maryland) on an Argus® Optical Mapping System. Contigs from Roche 454® and PacBio® sequencing were mapped onto the whole-genome restriction map using MapSolver® (v3.2.0). The presence of an extrachromosomal (i.e. plasmid) DNA element was confirmed using the Eckhardt gel technique[103], as modified by Hynes and McGregor[104]. The complete CF39S and pCF39S sequences were deposited into *Pseudomonas* genome database[45] and NCBI GenBank Accession numbers NZ_CP045916.1 and NZ_CP045917, respectively.

**Genome annotation.** Genomic DNA sequences for the reordered chromosome and its plasmid were input into Prokka v.1.12 (ref. [105]) using an *e*-value cut-off of 1e−30 for sequence similarity and the option to search for non-coding RNAs using Infernal + RFAM. *P. aeruginosa* PAO1 genome annotations in the Pseudomonas Genome Database[45] were transferred to *P. aeruginosa* CF39S. In short, a DIAMOND[106] search (*e*-value = 1e−12, query coverage = 100%, minimum percent identify = 95%, running in "sensitive" mode) using CF39S proteins against a PAO1 DIAMOND reference database of protein sequences (downloaded from http://pseudomonas.com) was performed. A reciprocal DIAMOND search against CF39S proteins was performed using the same stringent cutoffs and annotations for 1957 curated *P. aeruginosa* PAO1 proteins that had only one reciprocal best DIAMOND hit in *P. aeruginosa* CF39S were merged with the Prokka annotations. In order to identify putative genomic islands, the 6797918-bp chromosome was submitted to the IslandViewer 4 website (http://www.pathogenomics.sfu.ca/islandviewer/) using default settings[107]. In order to identify AMR genes, CF39S protein sequences were uploaded to the Comprehensive Antibiotic Resistance Database (CARD) website (Version 3.0.1) for input into the Resistance Gene Identifier (RGI) tool (Version 4.2.2) using "Perfect and Strict hits only" criteria and other default settings[108]. Finally, PSORTb v3.0 (ref. [109]) was used to predict subcellular localization of all CF39S proteins.

**Western blots.** Whole-cell lysates were prepared from *P. aeruginosa* strains grown on LB agar for 24 h at the indicated temperatures. Cells were collected using a polyester-tipped swab, suspended in PBS, and then collected by centrifugation at $10,000 × g$ for 5 min. Cell pellets were solubilized in Laemmli buffer by heating at 100 °C for 15 min. Afterward, total protein concentration was determined and 10 µg of protein from each sample was loaded onto a precast Any kD™ Mini-PROTEAN® TGX™ SDS-PAGE gel (BioRad®) along with the Precision Plus® Protein ladder (BioRad®). Electrophoresis was performed in Tris-glycine sodium dodecyl sulfate (TG-SDS, Amresco) running buffer for 45 min at 170 V on a Mini-PROTEAN® Tetra Vertical Electrophoresis Cell (BioRad®). Following SDS-PAGE, proteins were transferred onto a nitrocellulose membrane. Nitrocellulose membranes were cut to match the size of the gel and soaked in transfer buffer (containing 25 mM Tris base, 18.6 mM glycine, and 20% methanol). Afterward, proteins were transferred from the SDS-PAGE gel to the membrane using a Transblot® SD Semi-Dry Transfer Cell (BioRad®) run at 20 V for 45 min at room temperature.

Following protein transfer, the nitrocellulose membrane was blocked by incubating it in 5% skim milk in Tris-buffered saline–Tween 20 (TBS-T, which contained 25 mM Tris base, 137 mM NaCl, and 0.1% Tween 20, adjusted to pH 7.6) on an orbital shaker for 1.5 h. The membranes were then submersed in primary antibodies (α-PelC, α-PelD, or α-PslG) that were diluted 1:2500 in 1% skim milk in TBS-T buffer and left overnight at 4 °C on an orbital shaker. The following day, primary antibodies were removed, and the membrane was washed three times in TBS-T buffer. Subsequently, the membranes were incubated in secondary horseradish peroxidase-conjugated goat anti-rabbit IgG antibodies (ThermoFisher Scientific, product number 31460 and lot number TI208014) that were diluted 1:25,000 in TBS-T buffer for 1 h at room temperature on an orbital shaker. Lastly, the membrane was washed three times in TBS-T buffer. Proteins were visualized using the Clarity™ Western ECL substrate (BioRad®) and then the nitrocellulose membranes were imaged using a FluorchemQ gel documentation system (Protein Simple®). Western blot images were captured and analyzed using Alphaview (v3.4.0) (Proteinsimple). Original, unaltered Western blot images are presented in Supplementary Figs. 8, 9, and 10 for PelC, PelD, and PslG, respectively.

**Construction of expression vectors for protein production.** The *tdcA* ORF was cloned from *P. aeruginosa* CF39S gDNA using primer pair oJJH1692/oJJH1550. Primer oJJH1692 was tailed with an *attB1* and a tobacco etch virus (TEV) protease sequence, whereas oJJH1550 was tailed with an *attB2* sequence (Supplementary Table 3). The resulting PCR product was gel purified and recombined with pDONR221 using BP Clonase II. Clones were recovered on LB + Kn and the plasmid with the desired insert was identified by colony PCR and sequence-verified using M13F and M13R primers, yielding pJJH295. Subsequently, pJJH295 was recombined with the destination vectors pHNGWA and pHMGWA using LR Clonase II, thereby fusing TdcA to the hexahistidine-conjugated N-utilizing substance (His$_6$-NusA-TdcA) and hexahistidine-conjugated maltose binding protein (His$_6$-MBP-TdcA) solubility, respectively. These Gateway® reactions were then transformed into *E. coli* DH5α and clones were selected on LB + Cb agar. A single colony was picked from each agar plate, yielding pJJH319 and pJJH320 for the expression of recombinant His$_6$-MBP-TdcA and His$_6$-NusA-TdcA, respectively (Supplementary Table 2). Mutations to the putative catalytic site and truncation of the TdcA protein were generated via site-directed mutagenesis of pJJH319. Primer pair oTER47/oTER48 was used to introduce an E275A mutation in the putative catalytic site of TdcA, yielding pTER90 for the expression of His$_6$-MBP-TdcA$^{E275A}$ (Supplementary Table 2). Primer pair oTER170/oTER171 was used to introduce an M1* mutation in TdcA, yielding pTER91 for the expression of His$_6$-MBP (Supplementary Table 2). Primer pair oTER172/oTER173 was used to introduce the M167* mutation, yielding pTER92 for the expression of His$_6$-MBP-PAS$_{TdcA}$ (Supplementary Table 2).

The diguanylate cyclase, WspR, requires a phosphorylation signal for activation of cyclase activity; however, the V72D substitution leads to constitutive activation of WspR activity[70]. Here, a vector for producing recombinant WspR$^{V72D}$ was built by executing site-directed mutagenesis with primer pair oTER269/oTER270 on the protein expression vector pETduet::wspR/PA2133 (ref. [70]) to introduce the V72D mutation, generating the plasmid pETduet::wspR$^{V72D}$/PA2133 (Supplementary Table 2).

To generate a His$_6$-MBP-tagged fusion for the expression of recombinant *E. coli* DosP, the *dosP* ORF was cloned from *E. coli* K12 MG1655 gDNA using primer pair oJGR182/oJGR183 (Supplementary Table 3). The PCR product was recombined with pDONR221 using BP Clonase II. Clones were recovered on LB + Kn and the plasmid with the desired inserts was identified by colony PCR and sequence-verified using M13F and M13R primers, yielding pJGR10. Subsequently, pJGR10 was recombined with pHMGWA using LR Clonase II, thereby fusing DosP to 6×His-MBP. The LR Clonase II reaction was then transformed into *E. coli* DH5α and clones were selected on LB + Cb agar. A clone bearing the desired insert was identified by colony PCR and Sanger sequencing using primers oJJH1906 and oJGR161 (Supplementary Table 3), yielding pJGR11.

**Protein expression and purification.** The TdcA and DosP expression constructs were transformed into *E. coli* NiCo21 (DE3) (Supplementary Table 1) for protein expression. *E. coli* NiCo21 (DE3) transformants were incubated in SB + Cb (TdcA) or LB + Cb (DosP) at 37 °C and 250 r.p.m. until the culture reached an OD$_{600}$ of 0.6. Subsequently, the temperature was lowered to 18 °C and 0.5 mM isopropyl-β-D-thiogalactopyranoside (IPTG) was added to induce protein expression. Cells were harvested after 3 to 6 h by centrifugation (7000 × *g* for 20 min), followed by two wash steps with buffer A (50 mM Tris-HCl, 0.5 M NaCl, 10 mM imidazole, 2 mM DTT, pH 7.5), before the cell pellet was frozen at −80 °C. Cell pellets were cryogenically stored for up to 1 month.

Frozen cell pellets were thawed and suspended in 3 × v/m lysis buffer. Lysis buffer was composed of buffer A that had been amended with cOmplete Mini Protease Inhibitor Cocktail (EDTA-free, Roche) and 0.1 mM phenylmethanesulfonyl fluoride (PMSF). Cells were then lysed using a probe tip sonication device with 12 cycles consisting of 45 s sonication followed by 1 min of cooling on ice. Subsequently, the cell lysates were centrifuged twice for 20 min at 7000 × *g* at 4 °C, followed by centrifugation for 30 min at 20,000 × *g* at 4 °C, each time moving the supernatant to a new sterile container. These cleared cell lysates were then passed through a 0.45-μm Supor® filter (Pall).

His$_6$-NusA-TdcA, His$_6$-MBP-TdcA, and His$_6$-MBP-DosP protein purification was performed on an Äkta Pure FPLC (GE Health care). Cleared cell lysates were applied to a Ni-NTA column (HisTrap 5 ml FF, GE Health care). The column was first washed with 15 column volumes (CV) of 80% buffer B (50 mM Tris-HCl, 0.5 M NaCl, 10 mM imidazole, pH 7.3), and 20% buffer C (50 mM Tris-HCl, 0.5 M NaCl, 250 mM imidazole, pH 7.3), followed by 20 CV 70% buffer B and 30% buffer C. Protein was eluted in 2 ml fractions using first 25% buffer B and 75% buffer C, followed by 100% buffer C.

WspR$^{V72D}$ protein expression and purification were performed following the methods established by Huangyutitham and colleagues[70]. In brief, His$_6$-WspR$^{V72D}$ was expressed in *E. coli* Rosetta2 (DE3) transformed with pETduet::wspR$^{V72D}$/PA2133. This strain was grown in SB + Cb and incubated at 37 °C and 250 r.p.m. until the culture reached an OD$_{600}$ of 0.6. Temperature was lowered to 18 °C and 1.0 mM IPTG was added for induction of protein expression. Cells were harvested after 18 h by centrifugation at 7000 × *g* for 20 min at 4 °C. Cells were washed twice in 100 ml Buffer D (25 mM Tris, 0.3 M NaCl, 10 mM imidazole, pH 7.3) and pelleted a final time before being frozen at −80 °C. Storage of frozen cell pellets, lysis of cells, and removal of cellular debris from cell lysates by centrifugation was identical to the procedure described above for recombinant TdcA and DosP.

His$_6$-WspR$^{V72D}$ purification was performed on an Äkta Pure FPLC. Cleared cell lysates were applied to a HisPur Cobalt column (Thermo Scientific). The column was first washed with 15 CV of 95% buffer D and 5% buffer E (25 mM Tris, 0.3 M NaCl, 250 mM imidazole, 0.1 M arginine, pH 7.3). Next, 6×His-WspR$^{V72D}$ was eluted using a gradient elution step whereby the ratio of buffer E was increased from 10 to 100% (i.e. the imidazole concentration was increased from 25 to 250 mM). The fractions containing purified protein, which were eluted between 25 mM and 37.5 mM imidazole, were identified by on instrument A$_{280}$ measurements. These fractions were combined and concentrated using Amicon filter units with a 10-kD cut-off (EMD Millipore®) according to the manufacturer's directions. The concentrated protein was then diluted as required in buffer F (50 mM HEPES, 150 mM NaCl, 1 mM DTT, and 10% glycerol) and stored at −20 °C.

**UV–Vis spectroscopy of purified proteins.** Purified His$_6$-MBP-TdcA and His$_6$-MBP-DosP were diluted in protein buffer C to a concentration of 1 mg/ml for His$_6$-MBP-TdcA and 1.15 mg/ml for His$_6$-MBP-DosP, which was an empirically standardization based on the A$_{280}$ peak of each sample. The absorbance spectra for His$_6$-MBP-TdcA and His$_6$-MBP-DosP were collected in the range of 260–700 nm using a Lambda 35 UV/VIS Spectrometer with a Peltier device with SUPRASIL® quartz spectroscopy cells with a light path of 10 mm (Perkin-Elmer). The spectral scan was repeated six times for His$_6$-MBP-TdcA and His$_6$-MBP-DosP and each measurement was baseline corrected for buffer absorbance. Absorbance of hemin

and flavin mononucleotide (FMN) standards were measured in an identical manner except that these compounds were prepared in ethanol.

**C-di-GMP quantification using pyrophosphate assays.** Diguanylate cyclase activity of purified recombinant TdcA (His$_6$-MBP-TdcA or His$_6$-NusA-TdcA) and recombinant WspR (His$_6$-WspR) was measured using an EnzChek® Pyrophosphate Assay (Invitrogen) with modifications[70] also summarized here. High-purity guanosine 5′-triphosphate (GTP) was purchased from Affymetrix. Yeast pyrophosphatase was purchased from Roche. A standard curve was produced for each temperature using 50, 75, 100, and 125 μM pyrophosphate. We used 0.5 mM GTP for all measurements of diguanylate cyclase activity. Data were collected using a Lambda 35 UV/VIS Spectrometer with a Peltier device (Perkin-Elmer). Measurements were done using SUPRASIL® quartz spectroscopy cells with a light path of 10 mm (Perkin-Elmer).

**Nucleotide extractions and c-di-GMP quantification by mass spectrometry.** A culture was grown in 5 ml of LB broth at 37 °C at 250 r.p.m. The following day, the culture was diluted back to 1:100 in 55 ml of LB and grown to an OD$_{600}$ of 0.6 at 37 °C. Two 10 ml of culture aliquots were taken from the flask and passed through separate filters (0.45 μm, Millipore). One filter was used for determining cell count, while the other was used for nucleotide extractions. Both filters were placed on the same LB agar plate at the indicated temperature for 3 h. Following the incubation period, the first filter was washed in 2 ml of PBS and the cells in the PBS were used for viable cell counting. The second filter was processed as follows: microbial metabolites were extracted by transferring the second filter to a Petri dish (60 mm × 15 mm, Fisher Scientific) containing 2 ml of 4 °C 80% HPLC-grade methanol and allowed to incubate for 10 min; methanolic extracts were collected and the filter was washed with 1 ml of 4 °C 80% methanol; the methanolic extract and wash were combined in a 5-ml tube and insoluble debris was pelleted by centrifugation (30 min at 7000 × *g*); 2 ml of the supernatant was harvested and dried by using a speed vac and suspended in 200 μl of 50% methanol. The extracted nucleotides were stored at −80 °C until mass spectrometry analysis. After mass spectrometry analysis, the quantified c-di-GMP was normalized using the cell counts obtained for each sample.

Liquid chromatography mass spectrometry (LC-MS) analyses were conducted at the Calgary Metabolomics Research Facility (CMRF) on a Thermo Scientific Q Exactive™ Mass Spectrometer. Data were acquired in full scan mode (70–1000 *m/z*) at 140,000 resolving power. Metabolites were ionized via negative mode electrospray using the following source parameters: spray voltage −2000 V, capillary temperature 275 °C, auxiliary gas temperature 325 °C, sheath gas flow 25, auxiliary gas flow 10, sweep gas flow 2. Metabolites were separated using ultra-high performance liquid chromatography (UHPLC) following a reverse phase ion-pairing strategy adapted from Lu and colleagues[110]. Briefly, chromatographic separation of metabolites was achieved using a binary solvent system (Buffer A—10 mM tributylamine, 97% H$_2$O v/v, 3% MeOH v/v, pH adjusted to 7.5 using glacial acetic acid; Buffer B—acetonitrile) in conjunction with a Zorbax SB-C18 Rapid Resolution HT column (50 mm × 2.1 mm × 1.8-micron; Agilent Technologies). The following solvent gradient was employed: 0–1 min, 0% B; 1–4 min, 0–100% B; 4–5 min, 100% B; 5–5.5 min, 100–0% B; 5.5–8 min, 0% B. The c-di-GMP molecular assignment was established by high-resolution MS1 analysis, co-elution of the microbial metabolite with a commercial metabolite standard c-di-GMP (Sigma-Aldrich), and the assignment was confirmed via MS/MS fragmentation patterns observed using parallel reaction monitoring at 15 eV collision energy. LC–MS data were analyzed using MAVEN software[111] and c-di-GMP levels were quantified by MS1 peak intensities referenced to calibration curves prepared with c-di-GMP metabolite standards.

**LC-MS/MS quantification of c-di-GMP production by purified TdcA in vitro.** Diguanylate cyclase activity was also quantified for recombinant His$_6$-MBP-TdcA in vitro via LC–MS/MS measurements of c-di-GMP (as described above). Here, 10 μg of purified His$_6$-MBP-TdcA was added to 200 μl of reaction buffer (50 mM Tris base, 1 mM MgCl$_2$, pH 7.5). Reaction mixtures were equilibrated for 5 min in a heating block at the desired temperature prior to the addition of 1 μl of 100 mM GTP. The reactions were then quenched after 1 min by the addition of 200 μl of ice cold, HPLC grade, 100% MeOH in situ. Corrections for background c-di-GMP levels (carried over from protein production or resulting from low residual activities during quenching) were made by subtracting quantities of c-di-GMP from a negative control reaction, which was setup with 10 μg of purified His$_6$-MBP-TdcA in 200 μl of reaction buffer and handled identically to test reactions, except that the reactions were quenched with 100% MeOH, and 1 μl of 100 mM GTP was added post-quenching.

**Calculation of temperature coefficient ($Q_{10}$) values.** The temperature coefficient ($Q_{10}$) is the factor by which the rate of reaction increases for every 10 °C (or Kelvin) increase in temperature. The $Q_{10}$ value was calculated using the following equation[72]:

$$Q_{10} = \left(\frac{k_1}{k_2}\right)^{\left(\frac{10}{T_2 - T_1}\right)} \tag{2}$$

where $k_1$ is the reaction rate measured at temperature $T_1$, and $k_2$ is the reaction rate measured at temperature $T_2$.

**Bioinformatic surveys for TdcA homologs and molecular phylogenetics**. A search for TdcA protein homologs was executed using a BLASTP search of the NCBI non-redundant nucleotide database implemented within Geneious Prime v2020.1 (ref. [112]). Sequences were further analyzed using InterProScan 5 (ref. [113]). We set up a series of three filtering criteria to provide high-confidence predictions of other thermosensory diguanylate cyclases: (1) All sequences containing >2 putative domains, as identified using the Conserved Domain Database (CDD), were eliminated from further analysis; (2) Every sequence must possess a putative N-terminal PAS domain and a C-terminal a GGDEF domain; and (3) The query TdcA$_{Ps}$ sequence must align to both the PAS and GGDEF domains. This resulted in a list of 107 putative TdcA$_{Ps}$ homologs (Supplementary Data 1).

A phylogenetic tree was built using 39 of these high-confidence TdcA homologs in which genus and species could be unambiguously assigned to the genome. One representative sequence that had the highest identity with the query sequence was included for each species. MUSCLE[114] was used to make an alignment of these sequences, and WspR from *P. aeruginosa* PAO1 (NCBI Accession number AAG07089.1) was used as an outgroup. The phylogenetic tree was built using FastTree2 (ref. [115]) which was implemented within Geneious® Prime v2020.1 (ref. [112]).

Finally, the distribution of *tdcA* alleles in *P. aeruginosa* was investigated using a megaBLAST search with *P. aeruginosa* CF39S *tdcA* as the query sequence for the NCBI non-redundant database. This query included all complete and draft genomes, as well as plasmids and bacteriophages, which were restricted to taxon *P. aeruginosa* (taxID 287). A total of 119 sequences were identified. These sequences were exported into Geneious® v8.1.9 and a Geneious alignment was performed on the sequences. Using this alignment, a phylogenetic tree was generated from a Jukes-Cantor genetic distance model using neighbor-joining and no outgroup (Supplementary Fig. 2). The geographical distribution of *tdcA* alleles was identified by manually looking at sample metadata available through the NCBI database (Supplementary Data 1).

***Galleria mellonella* infection model**. *Galleria mellonella* larvae were used as a model[116,117] to assess a change in *P. aeruginosa* virulence with respect to temperature. *G. mellonella* larvae were obtained in bulk from The Worm Lady (McGregor, Ontario, CA) and stored at 16 °C until used. Bacterial cultures were grown in LB for 16 h at 37 °C and 275 r.p.m. Cultures were standardized in 0.85% saline using a 0.5 McFarland Standard (DensiCHEK® Plus, Biomerieux, USA) to obtain a cell density of $1.5 \times 10^5$ CFU/µl, which was then diluted in PBS to 10 CFU/µl. For infection, the abdomen of each larva was cleaned using a cotton swab and 70% ethanol and placed on an ice pack to minimize their motility. Next, 10 µl of diluted inoculum (100 CFU/larvae) was injected using a Hamilton syringe (Cole-Palmer) in the left lower abdominal proleg; 10 µl of sterile PBS was injected as control. A group of ten larvae was used for each strain and condition. Infected larvae were transferred to a clean Petri dish lined with filter paper, incubated at either 25 °C or 37 °C, and scored for survival every 24 h for 72 h. At least four biological replicates were used and survival curves were plotted using Graphpad Prism® v7.02.

***Caenorhabditis elegans* infection model**. Nematode maintenance and implementation of survival assays were performed as according to established procedures[117], which are summarized here. *C. elegans* N2 was maintained at 15 °C on 60 mm plates of Nematode Growth Medium (NGM) (containing 0.25% peptone) spread with *E. coli* OP50 as a food source[117]. For the survival assay, single colonies of each of *P. aeruginosa* strains (CF39, CF39s, JJH905, and TER77) were inoculated into 3 ml of LB broth and grown for 14–16 h at 37 °C after which 10 µl of the culture was spread onto 35 mm plates of modified NGM (containing 0.35% peptone), incubated at 37 °C for 24 h, and then at 25 °C for 24 h prior to seeding with ~30 L4 stage nematodes. The survival of the nematodes was monitored over time at 25 °C with live/dead scoring of nematodes twice daily. Nematodes were transferred to fresh assay plates every 2 days to separate subjects from progeny. Nematodes that died because of crawling off the plate were censored from the analysis. Experiments were conducted in triplicate and repeated at least three times. For each survival assay, nematode survival was calculated by the Kaplan–Meier method and survival differences were tested for significance using the log-rank test with Graphpad Prism® v7.02.

**Reporting summary**. Further information on research design is available in the Nature Research Reporting Summary linked to this article.

## Data availability

Finished CF39S and pCF39S sequences were deposited in the *Pseudomonas* genome database[45] (www.pseudomonas.com), and are also available through NCBI GenBank with accession numbers NZ_CP045916.1 and NZ_CP045917, respectively. Unedited, original images and replicates of Western blots shown in Fig. 2 appear in Supplementary Figs. 8–10. Sequences and metadata used to generate Supplementary Figs. 2 and 3 are provided in Supplementary Data 1. All bacterial strains and plasmids are available from the corresponding author on request. Source data are provided with this paper.

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

## Acknowledgements

The authors thank Lynne Howell for providing anti-PelC, anti-PelD, and anti-PslG antibodies. T.E.R. and J.D.R. have been supported by a Queen Elizabeth II Scholarship, and H.A. by an Eyes-High Postdoctoral Scholar Award from the University of Calgary. J. J.H. is supported by the Leader's Opportunity Fund of the Canada Foundation for Innovation (CFI), and a Project Scheme Grant and Canada Research Chair from the Canadian Institutes for Health Research (CIHR).

## Author contributions

H.A. and T.E.R. contributed equally to this work. J.J.H. conceived the project. B.Y.G., J.M., and J.J.H. designed the studies. H.A., T.E.R., and J.J.H. wrote the manuscript. H.A., T.E.R., E.A., J.D.R., K.L., L.A.S., and J.J.H. constructed the strains. H.A., T.E.R., E.A., F.L., K.L., N.F., and J.J.H. carried out the motility assays, biofilm assays, and/or photography. H.A., F.L., and N.F. carried out the gene expression measurements. H.A., T.E.R., L.A.S., W.K., G.L.W., F.S.L.B., R.E.B., M.R.P., M.F.H., and J.J.H. performed the genome sequencing and bioinformatic analyses. L.K.J., R.G., and I.A.L. performed the LC–MS/MS measurements. T.E.R., H.A., N.M.G., B.S.T., T.M.L.W., J.G., J.M., and J.J.H. carried out the protein production and/or enzyme assays. Y.L., E.G., M.S., M.S.S., A.K., and A.K.B. performed an analysis of virulence and executed the *Caenorhabditis elegans* and *Galleria mellonella* infection assays. All authors revised and provided feedback on the manuscript.

## Competing interests

H.A., T.E.R., M.R.P., J.M., and J.J.H. have filed patents for the use of heat-activated gene expression and synthetic proteins in biotechnology. All other authors declare no competing interests.
