## [Peer Review File · Nature Communications]

Editorial Note: This manuscript has been previously reviewed at another journal that is not operating a transparent peer review scheme. This document only contains reviewer comments and rebuttal letters for versions considered at *Nature Communications* .

REVIEWERS' COMMENTS

Reviewer #1 (Remarks to the Author):

This is a manuscript that I reviewed earlier. The authors should be commended for the extensive work they have put in this revised version and I do not have any further comments.

Reviewer #2 (Remarks to the Author):

This manuscript has been much improved in response to the reviewers comments, to the extend that it is now submitted as an article and includes a wide range of additional data/analysis.

While the actual mechanism of thermosensing remains unclear, the collective amount of data presented now presents a compelling picture of the physiological role these PAS domains can play. I support publication.

Reviewer #3 (Remarks to the Author):

The authors have responded adequately to all my queries. The revised manuscript represents a very solid and significant piece of work that will have substantial impact in the field. I would like to compliment the authors on their important accomplishment.

Response to reviewer comments (Almblad, Randall and colleagues, 2020)

- *Thank you for the positive feedback on our work. As requested, we have listed comments from the peer-reviewers below and have provided our responses point-by-point and highlighted them in italics. Only minor modifications to the manuscript have been made as itemized in the accompanying editorial checklist. Also, please find attached a detailed Nature Reporting summary as requested.*

Reviewer comments

Reviewer #1

This is a manuscript that I reviewed earlier. The authors should be commended for the extensive work they have put in this revised version and I do not have any further comments.

- *Thank you for your feedback and the commendation. We worked our hardest to address all the feedback that was previously provided on our manuscript.*

Reviewer #2

This manuscript has been much improved in response to the reviewers comments, to the extent that it is now submitted as an article and includes a wide range of additional data/analysis. While the actual mechanism of thermosensing remains unclear, the collective amount of data presented now presents a compelling picture of the physiological role these PAS domains can play. I support publication.

- *Thank you for supporting publication. We are actively working on elucidating the mechanism of thermal sensing and are looking forward to figuring it out (we note that an analogous mechanism for neuronal thermoTRPs remains unknown too). We agree that the thermoPAS domain is likely to play a role in many aspects of signal transduction.*

Reviewer #3

The authors have responded adequately to all my queries. The revised manuscript represents a very solid and significant piece of work that will have substantial impact in the field. I would like to compliment the authors on their important accomplishment.

- *Thanks for your positive feedback and compliments!*